# Sparse Topology Pairwise Scoring for Large-Scale Multi-Agent Reinforcement Learning

## Abstract

In multi-agent reinforcement learning (MARL), the problem of partial observability and stochasticity can be alleviated by enabling agents to access additional information about others through communication. However, in large-scale settings, communication among agents leads to a quadratic increase in the number of pairwise links, resulting in excessive bandwidth consumption and memory bottlenecks. Previous studies primarily aimed to solve this problem through learning globally optimal communication graphs, but such designs inevitably incur rapidly escalating complexity as the agent population increases. In this work, we propose a scalable communication scheme for large-scale MARL, termed *Sparse tOpology Pairwise Scoring* (SOPS). We hypothesize that leveraging pairwise relations among agents over an efficient backbone topology can enhance cooperative policies, and we adopt an exponential graph as a scalable backbone topology with a small diameter. Based on this backbone, we learn a probabilistic subgraph distribution parameterized by a pairwise scoring network that adaptively incorporates agent states and edge-type embeddings. To enable gradient-based optimization through discrete subgraph sampling, we employ Gumbel-Sigmoid reparameterization, whose differentiable nature allows the entire framework to be trained in an end-to-end manner. Overall, SOPS maintains high communication efficiency while adapting dynamically to task requirements and temporal variations. Evaluation results show that SOPS significantly outperforms existing state-of-the-art methods across cooperative benchmarks of diverse scales, consistently achieving higher rewards and faster convergence. SOPS also exhibits robust zero-shot transfer capabilities, enabling a model trained on a smaller scale to effectively apply to larger-scale scenarios. [1]

## 1 Introduction

Cooperative multi-agent reinforcement learning (MARL) has achieved significant advancements in a variety of complex decision-making applications, including complex virtual games (Barambones et al., 2022), simulated industrial control (Wang et al., 2022b; Leroy et al., 2023), autonomous driving (Dinneweth et al., 2022) and robotic planning (Zhang et al., 2021). These scenarios are often constrained by non-stationarity and partial observability (Omidshafiei et al., 2017). A widely adopted solution is the centralized training with decentralized execution (CTDE) paradigm, which leverages global information to train value functions or critics while constraining each agent to local observations at test time (Kraemer & Banerjee, 2016; Lowe et al., 2017). Building upon the CTDE framework, explicit inter-agent communication facilitates improved coordination by allowing agents to exchange mission-critical information that extends beyond their local observations. Recent studies have demonstrated that communication significantly improves decision-making efficiency and robustness across diverse cooperative scenarios, including group-aware coordination (Duan et al., 2024), targeted and trusted message exchange (Sun et al., 2024), personalized communication (Meng & Tan, 2024), and intention-aware protocols inspired by the theory of mind (Wang et al., 2021). These advances show the crucial role of communication schemes in scaling MARL to more complex and dynamic environments.

---

[1]The code is available in the supplementary material.

Despite the steady progress, existing CTDE methods become increasingly impractical as the number of agents grows, falling into the curse of dimensionality (Oroojlooy & Hajinezhad, 2023). This issue has been widely recognized in recent work on large-population systems and many-agent MARL (Cui et al., 2022; He et al., 2022). They typically rely on global state or observation information, training with which usually leads to sharply declining performance and poor sample efficiency at large scales (Wang et al., 2023). We identify two key challenges in designing such scalable communication mechanisms for the CTDE paradigm: achieving communication efficiency at large scales (**scalability**) and ensuring the adaptability of methods to various scales (**adaptability**). First, even in decentralized settings, dense or all-to-all communication patterns generate excessive message traffic and activation storage, quickly exceeding available bandwidth and GPU memory (Das et al., 2019; Iqbal & Sha, 2019a), regardless of centralized or decentralized settings. Second, learned communication structures tend to be fragile, exhibiting instability during training or poor generalization when applied to different agent populations or dynamic environments (Du et al., 2021; Hu et al., 2021).

Recently, some studies have attempted to scale MARL systems with reduced communication, partially addressing these challenges. For example, Chiun et al. (2025) enhance large-scale multi-robot exploration by pruning Graph Attention Networks and actions based on frontier-based information gain to avoid all-to-all communication. GTDE (Li et al., 2025a) learns adaptive groups for sharing information during training, stabilizing training through intra-group cross-agent information and mitigating non-stationarity. However, both methods start off with a full communication topology, and the resulting communication bandwidth is not properly bounded. Moreover, due to the limitations imposed by its network architecture design, GTDE's grouping strategy, learned from an agent population, cannot be seamlessly applied to another population scale. Li et al. (2025b) introduce a rule-based exponential communication topology, whose small-diameter communication is proven with bounded bandwidth for various scales. However, this fixed topology has difficulty in adapting communication links to task dynamics, thereby reducing knowledge exchange efficiency. Complementary to topology-based approach, large-scale MARL has also been explored via mean-field and attention-based approximations, such as GAT-MF for very large-scale MARL (Hao et al., 2023) and mix-attention approximations for homogeneous large-scale systems (Shike et al., 2023).

In this work, we propose *Sparse tOpology Pairwise Scoring* (SOPS), a scalable, sparse communication mechanism that separates global reachability from task-adaptive selection in the CTDE paradigm. Concretely, we instantiate an exponential graph as a small-diameter backbone, which guarantees fast multi-hop reachability under tight bandwidth and keeps the communication cost near-linear in the number of agents. On top of this backbone, a lightweight pairwise-scoring module selects a few peers per agent at each timestep via Gumbel-based sampling (Jang et al., 2017) with a linearly annealed temperature, yielding a task-adaptive, time-varying communication graph. Messages are aggregated with cross-attention blocks to accumulate multi-hop information, and simple auxiliary objectives are used to ground the messages during training. SOPS is plug-and-play with common value-based learners (e.g., IQL (Kostrikov et al., 2021)/QMIX (Samvelyan et al., 2019)) and executes fully decentralized without a proxy. Through extensive experiments in various cooperative MARL scenarios of various scales, SOPS outperforms other state-of-the-art (SOTA) methods by consistently achieving higher returns and faster convergence. It also exhibits robust zero-shot transfer ability to larger agent populations. Our contributions can be summarized as follows:

- SOPS is efficient and scalable. Its lightweight pairwise scoring enables budget-aware dynamic links in the exponential topology-based communication for large-scale MARL.
- SOPS is adaptable to various scales. Gumbel sampling and linear annealing allow end-to-end training and deployment across varying population sizes without topology redesign or population-specific retraining.
- SOPS demonstrates SOTA performance in various large-scale cooperative scenarios. It achieves higher returns and faster convergence, and has robust zero-shot transfer ability.

## 2 RELATED WORK

**Multi-agent Cooperation Paradigms.** The study of cooperation in MARL has evolved rapidly, with a variety of paradigms proposed to enable effective coordination in partially observable environments (Foerster et al., 2016). Central to this progress is the CTDE paradigm, which leverages global state information during training while enabling agents to act independently based on local

observations at test time (Oliehoek et al., 2016; Lowe et al., 2017). Extensions of CTDE, such as value decomposition methods (Sunehag et al., 2017; Samvelyan et al., 2019; Wang et al., 2020), aim to decompose the global team reward into local value functions, improving scalability and interpretability. Beyond reward decomposition, advances in credit assignment and counterfactual reasoning (Foerster et al., 2018) align individual behavior with team objectives under partial observability. Other lines of work investigate hierarchical cooperation frameworks (Yang et al., 2019; Iqbal et al., 2022), where agents operate under high-level coordination strategies, enabling task abstraction and modular training. Despite these developments, challenges remain in generalization and scalability to large agent populations. Kontogiannis et al. (2025) propose SMPE2, a state modeling framework that enables agents to infer agent-specific belief representations under partial observability and leverages them for adversarial exploration to enhance policy learning.

**Communication Learning and Topology Design.** Effective communication among agents plays a crucial role in improving coordination, especially in partially observable or dynamic environments. Early work introduced differentiable communication channels, enabling agents to learn end-to-end message exchange during training (Sukhbaatar et al., 2016; Das et al., 2019). Later methods leverage attention mechanisms to allow agents to selectively aggregate information from relevant peers, improving efficiency and interpretability (Iqbal & Sha, 2019b). Beyond continuous message passing, discrete protocols and emergent language have been explored to co-evolve communication and task policies (Eccles et al., 2019; Chafii et al., 2023). Representative topology-aware approaches include GTDE (Li et al., 2025a) and ExpoComm (Li et al., 2025b). GTDE learns grouping structures on top of a dense communication graph so that agents within the same group can share information, which stabilizes training but keeps instantaneous bandwidth unbounded and ties the learned grouping to the training population size, making it difficult to perform zero-shot resizing. ExpoComm instead hardcodes an exponential communication topology with provably small diameter and bounded bandwidth, but its rule-based neighbor sets are fixed and cannot adapt to task-specific or time-varying communication needs, which may lead to brittle or suboptimal routing. In contrast, SOPS keeps the exponential topology only as a sparse backbone for global reachability and formulates the selection of active links as a differentiable pairwise scoring problem, yielding a task-adaptive subgraph under strict bandwidth constraints; this perspective naturally connects communication topology design with recent advances in differentiable graph structure learning discussed next.

**Differentiable Graph Structure Learning.** Learning graph structures, particularly Directed Acyclic Graphs (DAGs), from data is a long-standing challenge in causal discovery, structured prediction, and communication learning. Traditional approaches often rely on discrete optimization or combinatorial search, which are computationally costly and incompatible with gradient-based training. Recent advances instead leverage differentiable continuous relaxations, i.e., continuous approximations of originally discrete sampling operations, allowing gradients to propagate through edge-selection processes and enabling end-to-end training in neural networks. A key breakthrough is the Gumbel-Softmax reparameterization (Jang et al., 2017), which relaxes categorical sampling into a continuous, differentiable form. Adding Gumbel noise to logits and applying a softmax with temperature enables gradients to flow through otherwise non-differentiable sampling operations. Building on these ideas, DP-DAG (Charpentier et al., 2022) extends differentiable sampling to globally constrained structures. It represents a DAG as a composition of a node permutation matrix and an upper-triangular edge matrix, sampling each with reparameterized distributions such as Gumbel-Sinkhorn, Gumbel-Top-k, and Gumbel-Softmax. By construction, the sampled adjacency matrix is acyclic, avoiding the need for explicit constraint penalties or cycle removal. This results in efficient, valid, and fully differentiable DAG learning.

## 3 PRELIMINARIES

### 3.1 COOPERATIVE MARL PROBLEMS FORMULATION

We model cooperative multi-agent reinforcement learning (MARL) tasks as a decentralized partially observable Markov decision process (Dec-POMDP) (Oliehoek et al., 2016). It is defined by the 8-tuple $\mathcal{M} = \langle \mathcal{I}, \mathcal{S}, \{\mathcal{A}_i\}_{i \in \mathcal{I}}, \mathcal{P}, \mathcal{R}, \{\Omega_i\}_{i \in \mathcal{I}}, \mathcal{O}, \gamma \rangle$, where $\mathcal{I} = \{1, \dots, n\}$ denotes the set of agents (with $n = |\mathcal{I}|$); $\mathcal{S}$ is the state space with global state $s^t \in \mathcal{S}$; $\mathcal{A}_i$ is agent $i$'s action space and the joint action is $a^t = (a_1^t, \dots, a_n^t) \in \prod_i \mathcal{A}_i$; the transition kernel is $\mathcal{P}(s^{t+1} \mid s^t, a^t)$;

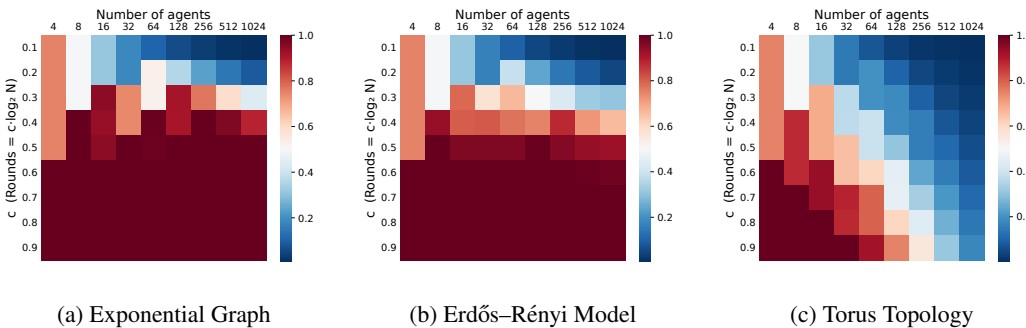

(a) Exponential Graph     (b) Erdős–Rényi Model     (c) Torus Topology

Figure 1: Comparison of log-step broadcast coverage under a unified per-round bandwidth. Columns give the number of agents, rows give a dimensionless time scale $c$, which sets the number of synchronous rounds $S = \lceil c \log_2 N \rceil$. Each cell shows the fraction of agents informed after $S$ rounds when every informed node may contact at most $K(N) = \lceil \log_2 N \rceil$ neighbors per round (if a node's degree is $< K$, it contacts all of its neighbors).

the team reward is $r^t = \mathcal{R}(s^t, a^t)$; $\Omega_i$ is agent $i$'s observation space and the joint observation is $o^t = (o_1^t, \dots, o_n^t) \in \prod_i \Omega_i$, generated by the observation function $\mathcal{O}(o^{t+1} \mid s^{t+1}, a^t)$. Each agent $i$ maintains a local history $\eta_i^t = (o_i^{1:t}, a_i^{1:t-1})$ and selects actions according to a decentralized policy $\pi_{\theta_i}(a_i^t \mid \eta_i^t)$. When communication is enabled, we consider a (possibly time-varying) directed communication graph $\mathcal{G}^t = (\mathcal{I}, \mathcal{E}^t)$ with neighborhood $\mathcal{N}_i^t = \{j \mid (j \to i) \in \mathcal{E}^t\}$. At each step, agent $i$ aggregates neighbors' previous-step messages to form the context $c_i^t = \mathrm{Agg}_i\{m_j^{t-1} : j \in \mathcal{N}_i^t\}$, then produces its own message $m_i^t = \mathrm{Msg}(h_i^t, c_i^t)$, conditioning their policies as $\pi_{\theta_i}(a_i^t \mid \eta_i^t, m_i^t)$. The objective is to learn decentralized policies $\pi = \{\pi_{\theta_i}\}_{i \in \mathcal{I}}$ that maximize the expected discounted return $J(\pi) = \mathbb{E}_{s^0 \sim \rho, \mathcal{P}, \mathcal{O}, \pi}\left[\sum_{t=0}^{T-1} \gamma^t r^t\right]$, where $\rho$ is the initial state distribution. This study focuses on the dynamic generation of $\mathcal{E}^t$ for each timestep $t$ to address efficient communication in large-scale cooperative MARL.

## 3.2 Exponential Topologies for Scalable Communication

Exponential graphs are a family of sparse communication topologies that enable efficient information dissemination among nodes (or agents) with low communication overhead (Ying et al., 2021; Chen et al., 2021; Li et al., 2025b). They are particularly well-suited for decentralized and multi-agent systems due to their small diameter and near-linear scaling in communication cost.

In an exponential graph, each node communicates with a fixed set of neighbors determined by powers of two. Specifically, each node $i$ (indexed from 0 to $N-1$) connects to nodes at distances $2^0, 2^1, \dots, 2^{\lfloor \log_2(N-1) \rfloor}$ modulo $N$. This results in each node having a degree of $\lceil \log_2 N \rceil$. The corresponding adjacency matrix $\mathcal{E}_{ij}^t \in \{0,1\}^{N \times N}$ at timestep $t$ is defined as:

$$\mathcal{E}_{ij}^t = \begin{cases} 1 & \text{if } \log_2((j-i) \bmod N) \in \mathbb{Z} \text{ or } i = j, \\ 0 & \text{otherwise.} \end{cases} \tag{1}$$

This topology ensures that the graph diameter is $\lceil \log_2(N-1) \rceil$, meaning any two nodes can exchange information within logarithmic time steps. The communication cost is $N \cdot \lfloor \log_2(N-1) \rfloor$, which is efficient compared to fully connected graphs.

To further highlight the advantages of static exponential connectivity, we conduct a comparative analysis against two representative alternatives: the Erdős–Rényi (ER) random graph and the Torus topology. The ER model is chosen as it has been shown to generate strong empirical performance and significantly more diverse connectivity patterns than Barabási-Albert or Watts-Strogatz models in multi-agent systems by Lou et al. (2024). In contrast, the Torus exemplifies a purely local, low-degree regular structure, which provides a natural worst-case comparison. As shown in Fig. 1, each heatmap cell records the average coverage fraction over multiple random sources (and graph resamples for ER). The results reveal a consistent ordering: the static exponential topology rapidly reaches near-complete coverage with a small $c$ and maintains scalability as $N$ grows; ER requires a

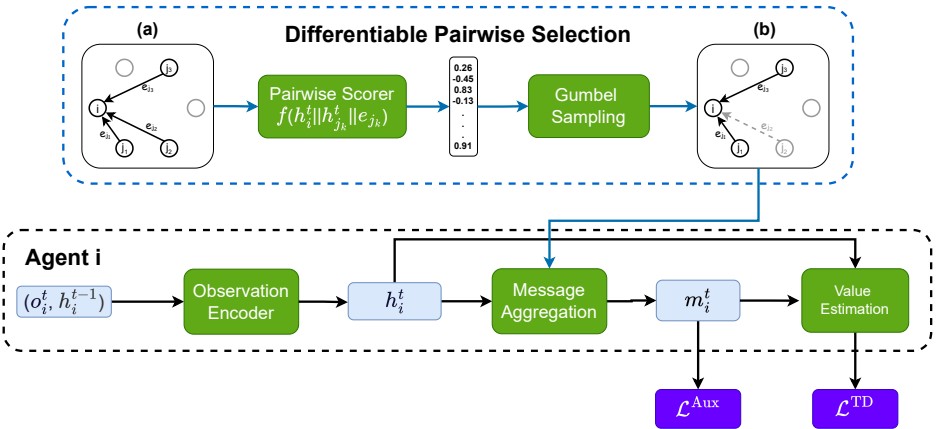

Figure 2: Overview of the SOPS architecture: (a) shows the exponential topology used as the backbone, and (b) illustrates the dynamically selected topology after pairwise selection.

larger $c$ and exhibits gradual degradation with scale; Torus remains the slowest, especially for large-scale systems. These findings confirm that the static exponential graph offers the most efficient and scalable backbone topology for communication in large-scale MARL.

## 4 METHODOLOGY

Motivated by the scalability and adaptability challenges discussed in the Introduction, our approach separates global reachability from task-adaptive communication selection under the CTDE framework. We construct a bandwidth-limited, small-diameter exponential backbone (see equation 1) to enable efficient multi-hop information dissemination at large scales. On top of this fixed backbone, we learn a sparse and time-varying communication subgraph that dynamically adjusts edge connections according to task requirements. The overall architecture is illustrated in Fig. 2. This design allows the model to adapt to varying dynamics without redefining the topology for different population sizes, ensuring both scalability and flexibility across diverse multi-agent settings.

### 4.1 SPARSE TOPOLOGY CONSTRAINED DIFFERENTIABLE PAIRWISE SELECTION

Optimizing a binary communication matrix $\mathcal{C}^t \in \{0,1\}^{N \times N}$ is difficult because it requires a differentiable mechanism that outputs discrete $0/1$ entries. A standard remedy is to treat $\mathcal{C}^t$ as a random matrix drawn from independent Bernoulli variables supported on the backbone, i.e., $\mathcal{C}_{ij}^t \sim \text{Ber}(\Theta_{ij}^t)$ with $\Theta^t \in [0,1]^{N \times N}$ and $\Theta^t$ masked by $\mathcal{E}^t$. To make this scalable, instead of optimizing $N^2$ free probabilities, we parameterize $\Theta^t$ via shared pairwise scoring on node embeddings only for backbone candidates. At step $t$, agent $i$'s candidate set is $\mathcal{N}_i^t = \{ j \mid \mathcal{E}_{ij}^t = 1 \}$. We first obtain node embeddings $z_i^t = \phi(h_i^t)$ from the local recurrent hidden state $h_i^t$, where $\phi$ is a shared projection that maps $h_i^t$ into a communication embedding space of dimension $d_z$. We then compute lightweight pairwise scores only on candidates:

$$\ell_{ij}^t = f(z_i^t \parallel z_j^t \parallel e_{ij}), \qquad \theta_{ij}^t = \sigma(\ell_{ij}^t), \qquad \Theta^t = \sigma(\mathcal{G}(Z^t)) \odot \mathcal{E}^t, \qquad (2)$$

where $e_{ij}$ is an edge-type embedding, $\sigma$ denotes the logistic function, $\theta$ is the parameterization, $Z^t = [z_1^t; \ldots; z_N^t] \in \mathbb{R}^{N \times d_z}$ denotes the matrix of per-agent communication embeddings at time $t$, and $\odot$ represents element-wise multiplication with the backbone mask. Conceptually, $\ell_{ij}^t$ is a sender-anchored score for the directed edge $i \to j$: for each sender $i$, the scorer evaluates all backbone candidates $j \in \mathcal{N}_i^t$. In our CTDE implementation these scores are computed centrally on the learner from the full tensor $Z^t$, but they are attached to outgoing edges from each sender and admit a straightforward decentralized implementation. Since scoring is performed only over backbone-defined candidates, both computation and memory costs avoid the explosive $O(N^2)$ growth of all-to-all interactions, laying the foundation for large-scale training.

Since gradients cannot pass through Bernoulli sampling directly, we adopt a Gumbel–sigmoid reparameterization with a straight-through estimator. We draw two i.i.d. Gumbel variables $g_{ij}^1, g_{ij}^2 \sim$ Gumbel$(0, 1)$ and use their difference to obtain logistic noise $\varepsilon = g_{ij}^1 - g_{ij}^2 \sim$ Logistic$(0, 1)$. Then we apply the reparameterization trick proposed by Maddison et al. (2017):

$$y_{ij}^t = \text{sigmoid}\left(\left(\log\left(\theta_{ij}^t/(1 - \theta_{ij}^t)\right) + \varepsilon\right)/\tau_t\right) \tag{3}$$

where $y_{ij}^t$ is the soft gate from the Gumbel relaxation and $\tau_t$ is the temperature at timestep t. In communication selection, deployment typically requires hard (0/1) links, so annealing is aligned with the final objective: begin with a higher temperature to obtain a smooth surrogate and sufficient exploration, then gradually reduce it so that the gates approach binary and stabilize (Jang et al., 2017). This makes an annealed schedule $(1.0 \rightarrow 0.3)$ typically yield smoother optimization early and a more stable topology later. Accordingly, we set $\tau_t = \max(\tau_{\min}, \tau_{\max} - \beta t)$, where $\beta$ is the decay rate and $t$ is the training step. Then we take a hard threshold at $0.5$ with a straight-through estimator, where $\mathbb{I}(\cdot)$ denotes the indicator function:

$$\mathcal{C}_{ij}^t = \mathbb{I}(y_{ij}^t > \tfrac{1}{2}) = y_{ij}^t + \text{stopgrad}(\mathbb{I}(y_{ij}^t > \tfrac{1}{2}) - y_{ij}^t). \tag{4}$$

Equivalently, since $\sigma$ is monotone, $\mathbb{I}(y_{ij}^t > \tfrac{1}{2}) = \mathbb{I}(\ell_{ij}^t + \varepsilon > 0)$, hence the hard sample follows $\mathcal{C}_{ij}^t \sim$ Ber$(\theta_{ij}^t)$ and is independent of $\tau_t$; the temperature only controls the smoothness of the surrogate and the gradient scale. At inference time, we use the deterministic rule $\mathcal{C}_{ij}^t = \mathbb{I}(\ell_{ij}^t > 0)$.

## 4.2 MESSAGE AGGREGATION AND DECISION MAKING.

At each time step $t$, agent $i$ generates message $m_i^t$ by combining its current hidden state $h_i^t$ with the aggregation of messages at the previous timestep from its selected in-neighbors $j$ at the previous step and generates the fused state $\hat{h}_i^t$:

$$m_i^t = \text{Agg}(\{\,\mathcal{C}_{ji}^t \cdot m_j^{t-1} : j \in \mathcal{N}_i^t\,\}), \qquad \hat{h}_i^t = \psi(h_i^t \,\|\, m_i^t). \tag{5}$$

Because maintaining all messages across multiple timesteps is memory-inefficient, we implement Agg as a lightweight scaled cross-attention mechanism. The fused state $\bar{h}_i^t$ then conditions the action-value network under CTDE.

## 4.3 TRAINING OBJECTIVE AND AUXILIARY TASK

Communication expands the policy space and can make optimization brittle when relying solely on the MARL objective, as prior work (Hu et al., 2024) notes. To help messages carry task-relevant global information, we follow the practice of using lightweight auxiliary grounding tasks that encourage local messages to accumulate and reflect multi-hop, multi-timestep information useful for decision making (Li et al., 2025b; Oord et al., 2018). When the global state is available during training, we supervise a small predictor to recover the current global state from local messages, encouraging messages to carry globally useful content:

$$\mathcal{L}_{\text{pred}}^{\text{Aux}}(\theta, \phi) = \mathbb{E}_{(s^t, o^t) \sim \mathcal{D}}[\,s^t - f(m_i^t; \phi)^2\,], \tag{6}$$

where $m_i^t$ is the message of agent $i$ at time $t$ (with $i$ sampled uniformly), and $f(\cdot; \phi)$ is a small learnable predictor used only for grounding (discarded at test time).

Otherwise, when the global state is unavailable, we adopt a contrastive InfoNCE objective that treats same-timestep messages from different agents as positives and messages outside a backbone-diameter temporal window (or from other agents/timesteps) as negatives:

$$\mathcal{L}_{\text{cont}}^{\text{Aux}}(\theta) = -\,\mathbb{E}_{i,j,t,t'}\left[\log \frac{\exp(g(m_i^t) \cdot g(m_j^t)/\kappa)}{\sum_{m \in \mathcal{M}} \exp(g(m_i^t) \cdot g(m)/\kappa)}\right], \tag{7}$$

where $i \sim$ Unif$\{1, \ldots, N\}$, $j \sim$ Unif$\{1, \ldots, N : j \neq i\}$, $g(\cdot)$ is an $\ell_2$-normalizing projection, $\kappa > 0$ is the temperature, and the set of negatives $\mathcal{M} = \{\,m_k^{t'} : k \in \{1, \ldots, N\},\ t' \notin [\,t - \text{diam}(\mathcal{G}^t),\ t + \text{diam}(\mathcal{G}^t)\,]\} \cup \{\,m_j^t\,\},\quad |\mathcal{M}| = M + 1$, with $M$ the number of negative pairs (a hyperparameter). The total training loss is:

$$\mathcal{L}^{tot}(\theta) = \mathcal{L}^{\text{TD}}(\theta) + \alpha \cdot \mathcal{L}^{\text{Aux}}(\cdot), \tag{8}$$

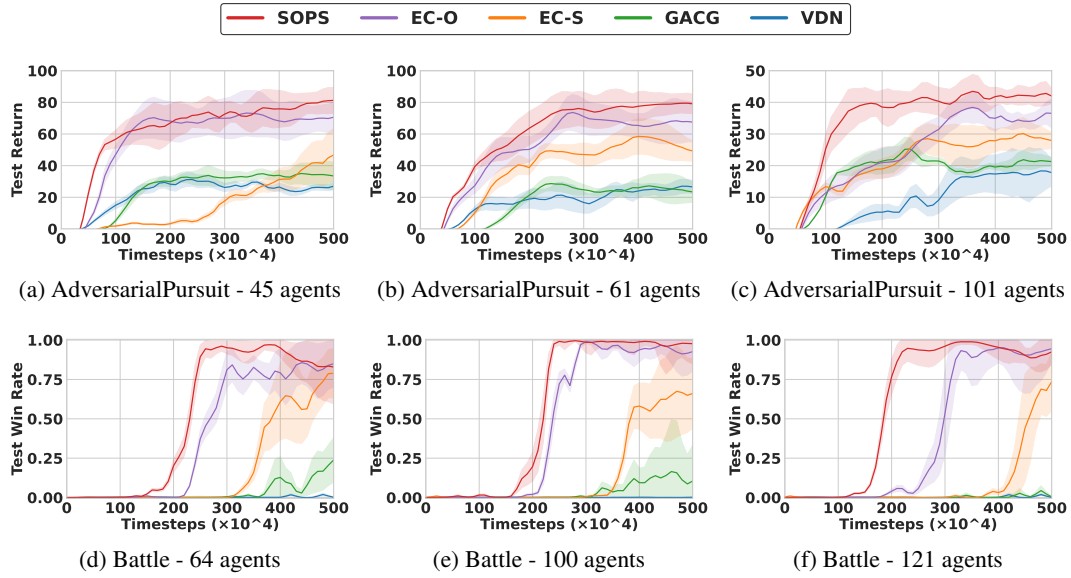

Figure 3: Performance comparison of SOPS and baselines on six MAgent scenarios. Return is the undiscounted cumulative reward per episode obtained by the learning (red) team during evaluation. Win rate is the proportion of evaluation episodes won by the learning team.

where $\mathcal{L}^{\text{Aux}}$ is equation 6 or equation 7 depending on whether the global state is available during training and $\alpha$ is the hyperparameter that balances the auxiliary term. The TD loss $\mathcal{L}^{\text{TD}}$ is defined according to Qmix (Samvelyan et al., 2019):

$$\mathcal{L}^{\text{TD}}(\theta) = \mathbb{E}_{(s^t, o^t, a^t, r^t, s^{t+1}, o^{t+1}) \sim \mathcal{D}} \left[ \left( y^{\text{tot}} - Q_{\text{tot}}(s^t, o^t, a^t; \theta) \right)^2 \right], \tag{9}$$

where $y^{\text{tot}} = r^t + \gamma \max_a Q_{\text{tot}}(s^{t+1}, o^{t+1}, a; \theta^-)$, and $\theta^-$ denotes the parameters of a periodically updated target network, as commonly employed in DQN (Van Hasselt et al., 2016).

## 5 EXPERIMENTS

In this section, we demonstrate through extensive experiments in large-scale settings that SOPS achieves higher normalized performance (return/win rate), faster convergence, and stronger zero-shot transfer capabilities compared to various baseline methods. And in ablations, we demonstrate the effectiveness of linear temperature annealing in Gumbel reparameterization for our method and show that it significantly outperforms alternative gradient estimation approaches in terms of training stability and final performance. Experiments were conducted on a server running Ubuntu 24.04 LTS, equipped with two Intel Xeon Platinum 8468 CPUs, 8 × NVIDIA A800 80GB PCIe GPUs, and 2 TiB of system memory. All SOPS experiments across scenarios were completed within 34 hours. Detailed hyperparameters for all experiments are provided in Appendix C.1.

### 5.1 SETUP

**Benchmark.** We experiment with two large-scale benchmarks for MARL: MAgent (Zheng et al., 2018; Terry et al., 2020) and Infrastructure Management Planning (IMP) from Leroy et al. (2023). Using these benchmarks, we conduct evaluations of SOPS and various baseline methods across 15 distinct scenarios, where the number of agents varies from 20 to 121. All experiments are averaged over three random seeds, and for the MAgent result plots, the shaded areas represent the 95% confidence interval. More additional details about the benchmarks are provided in Appendix C.2.

**Baselines.** We compare SOPS against four representative methods: (i) *VDN* (Sunehag et al., 2017): Traditional value decomposition method; no explicit inter-agent messaging ($K = 0$). (ii) *GACG* (Duan et al., 2024): Learns group assignments and performs sparse message passing within

Table 1: Performance comparison of SOPS and baselines on three IMP scenarios with $N = 50$ and $N = 100$ agents. Results are mean $\pm$ std of normalized discounted rewards relative to heuristic baselines. The best-performing method is in **bold**, the second best is underlined.

| Scenario | VDN | GACG | EC-O | EC-S | SOPS |
|---|---|---|---|---|---|
| | | | $N = 50$ | | |
| Uncorrelated | 23.43 ($\pm$6.27) | 25.51 ($\pm$4.50) | 27.68 ($\pm$4.89) | 28.21 ($\pm$4.40) | **29.10 ($\pm$4.21)** |
| Correlated | 18.04 ($\pm$8.78) | 39.50 ($\pm$7.55) | 42.10 ($\pm$3.44) | 44.02 ($\pm$5.47) | **45.59 ($\pm$5.03)** |
| OWF | 61.32 ($\pm$2.23) | 63.49 ($\pm$1.86) | 64.82 ($\pm$1.66) | 63.70 ($\pm$1.63) | **65.46 ($\pm$2.55)** |
| | | | $N = 100$ | | |
| Uncorrelated | 8.10 ($\pm$7.73) | 23.04 ($\pm$12.93) | 28.62 ($\pm$4.66) | 27.19 ($\pm$10.24) | **29.21 ($\pm$5.90)** |
| Correlated | -54.80 ($\pm$100.44) | 12.74 ($\pm$25.83) | 18.85 ($\pm$19.36) | 21.62 ($\pm$20.35) | **23.55 ($\pm$19.94)** |
| OWF | 64.88 ($\pm$2.21) | 65.60 ($\pm$0.78) | 65.96 ($\pm$0.50) | 63.76 ($\pm$1.21) | **66.40 ($\pm$0.48)** |

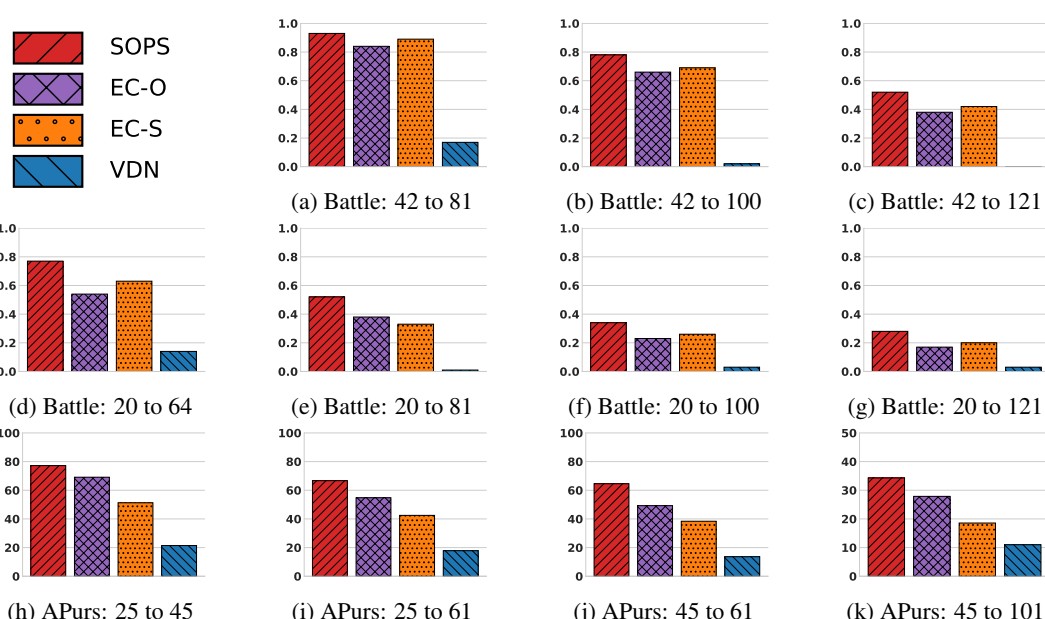

Figure 4: Performance comparison of zero-shot transfer performance on the MAgent environment. Subfigures (a) – (g) are under the *Battle* scenario, while (h) – (k) are under the *AdversarialPursuit* (APurs) scenario. Baseline GACG is excluded due to its network architecture being incompatible with this ability.

and across groups in an end-to-end manner. (iii) Two ExpoComm variants, *EC-S (static)* and *EC-O (one-peer)* (Li et al., 2025b): EC-S uses a static exponential topology as the communication backbone on a ring of $N$ agents. EC-O employs a single-neighbor exponential schedule. This variant retains the $\lceil \log_2(N - 1) \rceil$ diameter over time while keeping the instantaneous edge count linear ($N$ active edges per step). To ensure fairness, all methods are implemented in the same codebase and trained with a shared configuration whenever possible.

## 5.2 MAIN RESULTS

**MAgent.** Across both *AdversarialPursuit* and *Battle*, SOPS consistently learns faster and attains higher asymptotic performance than all baselines (Fig. 3). In all six settings with different agent populations, SOPS exhibits an early performance take-off and saturates near the task ceiling (returns or win rate), while EC-O is typically the runner-up but converges more slowly and with larger variance. EC-S improves more gradually and plateaus lower, suggesting that a denser static backbone is less sample-efficient at scale. GACG and VDN lag behind across populations, with VDN (no

Table 2: Plugging SOPS into different existing methods in the IMP scenario with 50 agents. Results are mean $\pm$ std of normalized discounted rewards relative to heuristic baselines. The best-performing method is in **bold**, the second best is underlined.

| Scenario | QMIX | QMIX + SOPS | QPLEX | QPLEX + SOPS | SHAQ | SHAQ + SOPS |
|---|---|---|---|---|---|---|
| Uncorrelated | 24.13 ($\pm$5.29) | **29.10** ($\pm$**4.21**) | 20.98 ($\pm$6.24) | 26.35 ($\pm$3.26) | -7.82 ($\pm$9.28) | 15.46 ($\pm$5.94) |
| Correlated | 18.04 ($\pm$7.37) | 45.59 ($\pm$5.03) | 20.10 ($\pm$4.82) | **46.26** ($\pm$**6.32**) | -14.88 ($\pm$19.52) | 32.41 ($\pm$10.90) |
| OWF | 61.38 ($\pm$4.26) | **65.46** ($\pm$**2.55**) | 60.74 ($\pm$3.18) | 62.93 ($\pm$2.36) | 49.90 ($\pm$13.79) | 58.38 ($\pm$7.69) |

communication) particularly struggling as the number of agents increases. To show that our method works well not just in extreme-scale scenarios, we also conduct experiments on smaller settings and observe consistent results, as detailed in Appendix C.3.1.

**IMP.** The quantitative summary on IMP (Tab. 1) reveals these trends: SOPS achieves the best mean normalized discounted reward in all tasks at both $N{=}50$ and $N{=}100$ ($N$ is the number of agents), with clear margins on *Correlated* and solid gains on *Uncorrelated* and *OWF*. EC-O/EC-S are competitive but consistently below SOPS, while VDN degrades sharply under correlation at $N{=}100$. Overall, SOPS maintains performance as $N$ increases, indicating superior scalability relative to baselines.

**Zero-shot transfer.** When transferring from smaller to larger populations without finetuning (Fig. 4), SOPS attains the highest win rates across all seven train→test pairs (e.g., $42 \rightarrow 81/100/121$). EC-O and EC-S retain certain transferability but drop more noticeably as the expansion factor grows; VDN performs poorly because of the lack of communication: trained at a small scale it learns only local heuristics, and transferring to larger populations leads to severe underfitting. GACG is excluded from the zero-shot transfer experiments because its fully-connected coordination graph and group structure are tied to a fixed agent set and are not directly reusable when the number or identities of agents change. These results demonstrate that SOPS preserves coordination under population scaling, aligning with its design goal of decoupling global reachability from task-adaptive sparse selection.

**Plug-and-play.** Our default implementation of SOPS is instantiated on top of QMIX. To verify that SOPS is not tied to a particular CTDE learner, we additionally evaluated it on the IMP benchmark with two additional value-decomposition methods, QPLEX (Wang et al., 2020) and SHAQ (Wang et al., 2022a), using the same training setup and hyper-parameters. The results are summarized in Tab. 2. Across all three IMP scenarios, attaching SOPS to each learner consistently improves performance over its backbone, and the best or second-best method in every scenario is always a "+ SOPS" variant. This indicates that SOPS brings complementary benefits that persist under different and more expressive value-decomposition learners.

## 5.3 ANALYSIS

We report additional quantitative and structural evaluations to complement the main results. The Area-under-the-curve (AUC) measurements for both MAgent and AdversarialPursuit, shown in Tab. 4 and Tab. 5, demonstrate that SOPS attains consistently higher cumulative performance under identical training budgets. The outcomes summarized in Fig. 3, together with the system-level statistics in Tab. 8, indicate that SOPS preserves coordination quality while maintaining moderate computational overhead as the number of agents increases. The learned-graph diagnostics in Tab. 6 and Tab. 7 further show that SOPS maintains sparse but high-coverage communication structures under zero-shot transfer, and that high-frequency edges remain largely stable across population changes. These analyses collectively reinforce the scalability and transfer properties of SOPS. Complete descriptions and visualizations are provided in Appendix C.3.

## 5.4 ABLATIONS

**Gumbel Temperature.** We study the effect of the Gumbel temperature on edge sampling, as shown in Fig. 5a. Our default linear annealing (SOPS) delivers the earliest take-off and the highest

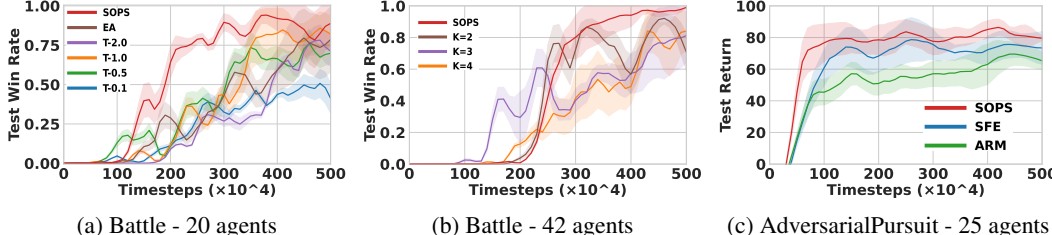

(a) Battle - 20 agents      (b) Battle - 42 agents      (c) AdversarialPursuit - 25 agents

Figure 5: Ablation studies on MAgent scenarios: (a) temperature (exponential annealing and $\tau = (0.1, 0.5, 1.0, 2.0)$), (b) edge-budget ($K = (2, 3, 4)$ communication links per agent), and (c) gradient-estimator variants (SFE, ARM).

final win rates at both 20- and 42-agent Battle scenarios. In contrast, exponential annealing (EA) cools too slowly early and too aggressively late, yielding delayed emergence of useful links and larger variance near convergence. Fixed temperatures underperform systematically: a high temperature (T-2.0) keeps gates overly soft and slows learning; a mid-value (T-1.0) improves but still lags; a lower value (T-0.5) learns faster but plateaus below SOPS; an overly low temperature (T-0.1) discretizes too early, hurting exploration and stability. These trends indicate that a smooth, schedule-driven hardening of edges is crucial for both sample efficiency and final performance, with linear cooling proving most robust across scales.

**Edge Budget.** Fig. 5b illustrates the benefit of learned neighbor selection over the exponential backbone with different edge budgets. In the 42-agent Battle setting, each agent has five candidate neighbors on the backbone. We remove the scorer and perform an ablation on the edge budget K, selecting edges based on Euclidean distance as the static feature. As shown in the figure, SOPS exhibits rapid improvement after the warm-up phase and converges stably to a high win rate. In contrast, the static baselines with K=2 and K=3 exhibit pronounced late-stage oscillations, indicating insufficient connectivity for stable coordination. The K=4 variant is more stable but learns slowly and fails to match SOPS's final performance. These results indicate that while the exponential backbone already provides a strong inductive bias, the main additional gains in both final performance and stability come from the learned, adaptive neighbor selection.

**Gradient Estimator.** Fig. 5c compares the performance of our default reparameterized Gumbel sampling approach against two representative score-function gradient estimators: the Score Function Estimator (SFE, also known as REINFORCE) and the Augment-REINFORCE-Merge (ARM) estimator (Yin & Zhou, 2019). SFE provides an unbiased but high-variance gradient estimate that requires careful baseline subtraction for stable training. ARM, specifically designed for binary variables, leverages symmetry properties to construct a lower-variance estimator, offering better performance than SFE but still lacking the gradient stability of reparameterization methods. While ARM successfully reduces variance and outperforms SFE, it still trails the reparameterized estimator in both peak performance and training stability. These results confirm that direct gradient flow through the sampling process, rather than score-function based estimation, is essential for achieving both high performance and training stability in discrete decision-making scenarios.

## 6 CONCLUSION

We present SOPS, a scalable communication mechanism for cooperative MARL that pairs a small-diameter exponential skeleton for rapid reachability with lightweight pairwise scoring to realize budget-aware sparse links; combined with Gumbel sampling and linear annealing, these discrete topology choices become end-to-end trainable and readily adaptable across population scales without redesign. Extensive experiments show that SOPS achieves higher returns and faster convergence, while exhibiting zero-shot transfer to larger agent populations. Ablations on temperature schedules and gradient estimators further confirm the importance of annealing and stable discrete optimization. With its plug-and-play compatibility with CTDE backbones, SOPS provides a practical path toward efficient, adaptable, and scalable communication in large-scale MARL.

## REPRODUCIBILITY STATEMENT

The source code for reproduction is available in the supplementary material. The benchmark environments and experimental configurations used in this work are also included in the supplementary material to ensure full reproducibility. Detailed hyperparameters for all experiments are provided in Appendix C.1.

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

## A    SOURCE CODE

The source code is available in the supplementary material.

## B    THE USE OF LARGE LANGUAGE MODELS (LLMS)

In preparing this manuscript, we used `ChatGPT` and `Qwen` as general-purpose writing aids to improve clarity, readability, and phrasing. They were not involved in any aspect of research conception, algorithm design, or coding. All scientific contributions, including technical ideas, methodology, and experimental design, were independently developed by the authors. We assume full responsibility for the content of this work.

## C    ADDITIONAL DETAILS OF EXPERIMENTS

### C.1    HYPERPARAMETERS

To ensure the fairness of comparison, we implement SOPS and all baselines within the same codebase, using shared hyperparameters except for those unique to specific methods. Following Li et al. (2025b), we use the default settings provided in the official implementations of MAgent (Zheng et al., 2018; Terry et al., 2020) and IMP (Leroy et al., 2023). For GACG (Duan et al., 2024), we use the officially recommended parameters: number of groups = 2 and trajectory length for group division = 10. For SOPS and two ExpoComm variants (Li et al., 2025b): loss coefficient $\alpha = 0.1$, loss temperature $\kappa = 0.07$ and number of negatives $M = 20$. Other common hyperparameters are given in Tab. 3.

Table 3: Common hyperparameters.

| Hyperparameter | MAgent | IMP |
|---|---|---|
| Hidden sizes | 64 | 64 |
| Discount factor $\gamma$ | 0.99 | 0.95 |
| Batch size | 32 | 64 |
| Replay buffer size | 2000 | 2000 |
| Number of environment steps | $5 \times 10^6$ | $2 \times 10^6$ |
| Epsilon anneal steps | $5 \times 10^5$ | $5 \times 10^3$ |
| Test interval steps | $5 \times 10^4$ | $2.5 \times 10^4$ |
| Number of test episode | 100 | 100 |

### C.2    BENCHMARKS AND SCENARIOS

**MAgent**    MAgent is a highly scalable many-agent reinforcement learning platform built on a large gridworld engine, supporting up to millions of agents on a single GPU through parameter sharing and agent ID embeddings (see Fig. 6). It provides flexible configuration for environments and agents, a reward description language enabling event-driven incentives, and an interactive renderer for real-time visualization. Agent actions are discrete — including move, turn, attack, or tag — and the underlying gridworld engine facilitates fast simulation of heterogeneous agent populations.

Two canonical scenarios within MAgent are AdversarialPursuit and Battle. In AdversarialPursuit, predators receive positive rewards upon successfully tagging prey, while prey incur penalties when tagged (a typical reward structure is defined in the platform's API). After training, predator agents typically develop local cooperative behaviors, forming dynamic enclosures to trap preys and accumulate cumulative rewards over successive timesteps.

In Battle, two large teams — each comprising hundreds of agents — compete on a shared map. Agents select from discrete actions (move, attack, idle) to cooperatively eliminate opposing team members. A team wins either by eliminating all enemy agents or by having more surviving agents at the end of the episode. Through self-play training, agents often evolve hybrid global–local strategies, such as coordinated encirclement or guerrilla-style skirmishing, reflecting emergent team-level coordination.

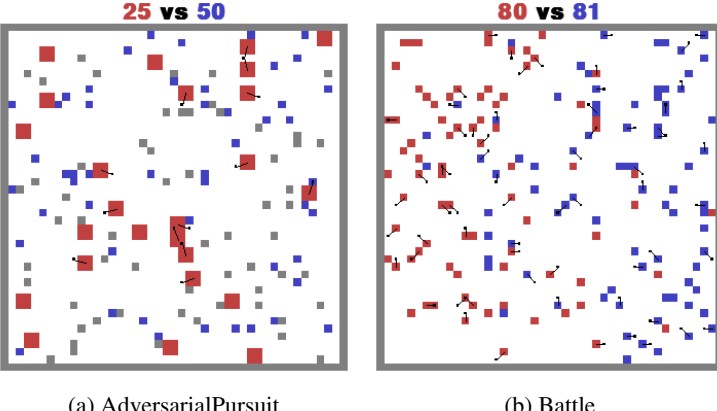

(a) AdversarialPursuit (b) Battle

Figure 6: AdversarialPursuit and Battle scenarios in the MAgent environment. In both scenarios, red squares denote agents controlled by the trained MARL policy, while blue squares represent agents governed by pretrained IDQN policies (Rashid et al., 2020)

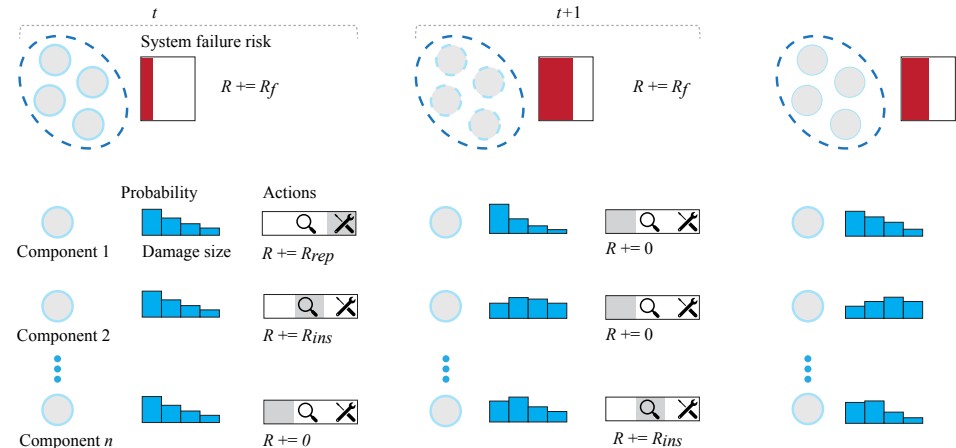

Figure 7: Infrastructure Management Planning (IMP) Environment. The system failure risk is modeled as a function of the probability distribution over each component's damage state. At every time step $t$, an agent—typically responsible for a single component—may choose to inspect or repair that component to regulate the failure risk. The IMP objective is to maximize the expected discounted return while trading off three (negative) reward terms: the system failure risk $R_f$, inspection costs $R_{\text{ins}}$, and repair costs $R_{\text{rep}}$. In the illustration, three components share the same damage probability at time $t$. If no action is taken, the damage probability follows a deterioration process.

**IMP** IMP is a cooperative multi-agent reinforcement learning benchmark for Infrastructure Management Planning, where each agent controls one system component and selects among three discrete actions: do nothing, inspect, or repair (see Fig. 7). Episodes are finite-horizon ($T = 20$–$30$), and the objective is to maximize the expected discounted return — balancing system failure risk against operation and maintenance (O&M) costs, with discount factor $\gamma = 0.95$. A configurable global campaign cost can be activated, imposing an additional timestep penalty for any inspection/repair action, thereby explicitly incentivizing coordinated scheduling across agents. The suite supports scaling to dozens or hundreds of agents via Gym/PettingZoo/PyMARL-compatible wrappers. Evaluation follows the authors' practice of normalizing returns relative to expert heuristic policies. Reward structure per step includes: (i) system failure risk $p_F^{\text{sys}}$ scaled by consequence, (ii) per-agent inspection/repair costs, and (iii) optional campaign cost — enabling evaluation of algorithms' ability to coordinate both spatially and temporally.

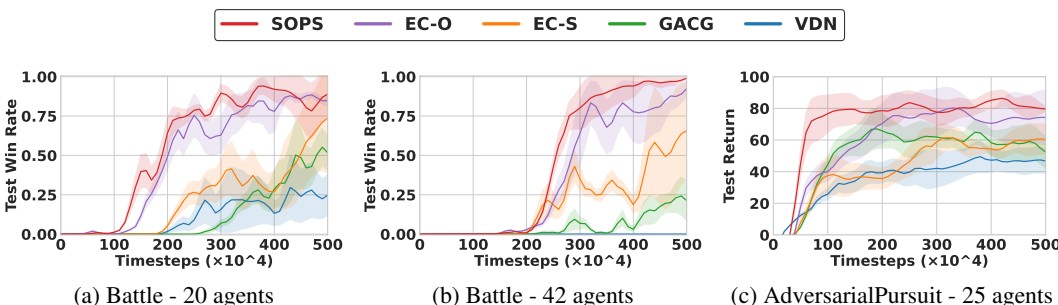

(a) Battle - 20 agents      (b) Battle - 42 agents      (c) AdversarialPursuit - 25 agents

Figure 8: Performance comparison of SOPS and baselines on three smaller-scale MAgent scenarios.

The benchmark includes three core scenarios: (1) Uncorrelated $k$-out-of-$n$: System fails if at least $(n - k + 1)$ components fail; damage states are represented by 30-bin probability vectors (last bin = failure), initialized independently per component. Observations concatenate normalized time with per-component damage distributions, testing coordination under independent deterioration. (2) Correlated $k$-out-of-$n$: Identical to above, but initial damage distributions are statistically correlated — inspecting one component reveals information about others. To mitigate partial observability, agents receive a shared correlation signal $\alpha_t$, updated from all inspection outcomes and appended to inputs, stressing cooperation under information coupling. (3) OWF (Offshore Wind Farm): Each turbine has three components (top/middle/mudline); mudline is unobservable/unrepairable, so two agents control each turbine (top + middle). Damage uses 60 bins, component models/costs vary by location, and turbines fail if any component fails; farm-level risk aggregates over turbines. This scenario emphasizes heterogeneous components and large-scale fleet management.

## C.3 ADDITIONAL RESULTS AND DISCUSSION

### C.3.1 FOR MAGENT BENCHMARK

Fig. 8 shows the performance of SOPS and baselines in three smaller-scale multi-agent scenarios. Across all settings, SOPS consistently achieves higher test win rates or returns compared to existing methods, with faster convergence and greater stability. These results demonstrate that SOPS maintains strong performance even in smaller-scale tasks.

To complement the main-figure analyses, we additionally report the area-under-curve (AUC) metrics for the evaluation curves of Battle and AdversarialPursuit. Unlike final performance alone, AUC provides a holistic measure of learning progress by integrating both convergence speed and asymptotic performance over the entire training horizon. For the top-3 method (EC-S, EC-O, and SOPS) and each scenario, we compute AUC separately for all three random seeds and report the mean and standard deviation. Tab. 4 and Tab. 5 show that SOPS consistently achieves higher AUC across both benchmarks, indicating superior sample efficiency and overall training stability. This further confirms the advantages of SOPS observed in Fig. 3 and Fig. 8, even in cases where confidence intervals visually overlap due to near-saturation of the evaluation metrics.

### C.3.2 ROBUSTNESS TO AGENT FAILURES AND TOPOLOGY DISRUPTIONS

Exponential graphs provide multi-scale, redundant connectivity—each agent links to $2^0, 2^1, \ldots$-offset neighbors—so the topology typically remains connected with near-logarithmic diameter even when some nodes or edges fail. Prior work similarly shows that exponential topologies preserve fast information mixing under random disruptions (Ying et al., 2021; Chen et al., 2021; Li et al., 2025b). On top of this backbone, SOPS does not depend on any specific edge: the pairwise scorer simply reallocates probability mass among the remaining candidates $N_i^t = \{j \mid E_{ij}^t = 1\}$, and cross-attention aggregation accumulates multi-hop information over time. Thus, the exponential skeleton offers inherent robustness, while task-adaptive sparse selection provides an additional mechanism to compensate for missing or unreliable links.

Although we do not perform targeted "agent failure" experiments, several of our existing results are consistent with robustness to topology perturbations. The broadcast-coverage study in Fig. 1

Table 4: Area-under-the-curve (AUC) of evaluation return curves for AdversarialPursuit scenario across three random seeds. AUC reflects cumulative learning progress under identical training budgets.

| Scenario | AdvPursuit-25 | AdvPursuit-45 | AdvPursuit-61 | AdvPursuit-101 |
|---|---|---|---|---|
| SOPS | 71.30 (±6.94) | 61.34 (±7.33) | 57.45 (±7.30) | 31.65 (±3.34) |
| EC-O | 58.67 (±7.95) | 56.17 (±9.17) | 49.36 (±7.54) | 21.61 (±3.33) |
| EC-S | 41.40 (±6.17) | 13.41 (±3.44) | 35.77 (±5.50) | 18.82 (±2.95) |

Table 5: Area-under-the-curve (AUC) of evaluation win-rate curves for Battle scenario across three random seeds. AUC jointly captures both convergence speed and asymptotic performance. Higher values indicate better overall learning efficiency.

| Scenario | Battle-20 | Battle-42 | Battle-64 | Battle-100 | Battle-121 |
|---|---|---|---|---|---|
| SOPS | 0.56 (±0.05) | 0.46 (±0.04) | 0.51 (±0.05) | 0.57 (±0.01) | 0.60 (±0.05) |
| EC-O | 0.49 (±0.04) | 0.37 (±0.07) | 0.39 (±0.04) | 0.49 (±0.03) | 0.39 (±0.05) |
| EC-S | 0.22 (±0.03) | 0.20 (±0.07) | 0.18 (±0.06) | 0.16 (±0.05) | 0.08 (±0.01) |

shows that exponential graphs maintain fast dissemination even when connectivity is effectively randomized or sparsified, outperforming Erdős–Rényi and Torus alternatives under matched budgets. Moreover, the zero-shot transfer experiments in Fig. 4 can be viewed as a strong form of structural perturbation: the agent population, and thus the communication graph, is resized between training and deployment without finetuning. SOPS preserves high coordination quality under these topology changes, suggesting that the learned communication scheme is not fragile to moderate structural variations.

### C.3.3 LIMITATIONS AND FUTURE EXTENSIONS

Our current analysis and experiments focus on settings where the exponential backbone can be instantiated over all active agents and remains largely intact during training and evaluation. We do not explicitly simulate adversarial or correlated failure patterns (e.g., removal of contiguous segments on the ring) or hard communication outages at scale. While the structural redundancy of exponential graphs and the task-adaptive selection in SOPS suggest a degree of inherent robustness, a systematic empirical study with backbone-edge dropout, synthetic agent failures, or failure-aware signals is left for future work.

In addition, our experimental suite primarily uses benchmarks with homogeneous agents (or weak heterogeneity), as is standard in large-scale MARL (e.g., MAgent, IMP). Algorithmically, SOPS itself does not rely on agent homogeneity: the exponential topology is defined over agent indices, and the pairwise scorer operates on learned embeddings that can naturally encode agent type, capability, or role. Nevertheless, we do not yet provide systematic evaluations in strongly heterogeneous, role-rich environments where highly targeted, type-aware communication may be especially beneficial. Extending SOPS with explicit type-aware edge features and assessing its performance in large-scale heterogeneous scenarios is a natural and important direction for future work.

### C.3.4 LEARNED-GRAPH DIAGNOSTICS

To better understand why SOPS transfers well under population resizing, we instrument the learned sparse communication graph during test-time rollouts, both in-distribution and under zero-shot transfer, and report structural diagnostics in Tab. 6–7. Tab. 6 summarizes graph sparsity and multi-hop coverage. Across all train→test pairs, the average out-degree remains in a narrow range around 3–4.5, substantially smaller than the exponential candidate set size. This indicates that SOPS consistently maintains a sparse per-agent communication load even after resizing. The coverage metrics in Tab. 6 further show that this sparse graph still provides fast information propagation. In moderate resize regimes such as Battle $42 \rightarrow 81$ and AdversarialPursuit $25 \rightarrow 45$, coverage within 3–4 hops quickly approaches 0.9 or higher, implying a small effective diameter and near-global reachability in just a few communication rounds. For more extreme enlargements (e.g., Battle $20 \rightarrow 100/121$

Table 6: Graph sparsity and multi-hop coverage across zero-shot transfer settings. For each train→test pair, we report the average out-degree and the average fraction of reachable agent pairs within $h$ hops (coverage@$h$) measured at test time.

| Setting (train → test) | Avg. out-degree | Coverage@2-hop | Coverage@3-hop | Coverage@4-hop |
|---|---|---|---|---|
| Battle 20→64 | 3.21 | 0.34 | 0.61 | 0.88 |
| Battle 20→81 | 3.35 | 0.20 | 0.45 | 0.63 |
| Battle 20→100 | 3.43 | 0.14 | 0.33 | 0.42 |
| Battle 42→81 | 4.32 | 0.39 | 0.68 | 0.90 |
| Battle 42→100 | 4.46 | 0.27 | 0.55 | 0.72 |
| Battle 42→121 | 4.58 | 0.18 | 0.40 | 0.53 |
| AdversarialPursuit 25→45 | 3.33 | 0.42 | 0.66 | 0.93 |
| AdversarialPursuit 25→61 | 3.49 | 0.35 | 0.57 | 0.72 |
| AdversarialPursuit 45→61 | 4.04 | 0.32 | 0.59 | 0.75 |
| AdversarialPursuit 45→101 | 4.35 | 0.24 | 0.46 | 0.62 |

Table 7: Edge persistence across zero-shot transfer settings. For each train→test pair, we report the Pearson correlation between edge activation frequencies and the Jaccard overlap between the top-$k\%$ most frequently used edges.

| Setting (train → test) | Pearson $\rho$ | Jaccard@5% | Jaccard@10% | Jaccard@20% |
|---|---|---|---|---|
| Battle 20→64 | 0.93 | 0.39 | 0.81 | 0.60 |
| Battle 20→81 | 0.85 | 0.44 | 0.76 | 0.55 |
| Battle 20→100 | 0.79 | 0.36 | 0.67 | 0.48 |
| Battle 42→81 | 0.95 | 0.47 | 0.88 | 0.68 |
| Battle 42→100 | 0.88 | 0.35 | 0.82 | 0.59 |
| Battle 42→121 | 0.85 | 0.40 | 0.74 | 0.50 |
| AdversarialPursuit 25→45 | 0.92 | 0.45 | 0.78 | 0.65 |
| AdversarialPursuit 25→61 | 0.88 | 0.37 | 0.70 | 0.50 |
| AdversarialPursuit 45→61 | 0.95 | 0.42 | 0.85 | 0.77 |
| AdversarialPursuit 45→101 | 0.87 | 0.28 | 0.76 | 0.52 |

or AdversarialPursuit $45 \to 101$), coverage@3 and coverage@4 decrease but remain substantially above what would be expected from random sparse graphs, mirroring the larger but still moderate drop in zero-shot performance in Fig. 4.

Tab. 7 complements these structural statistics with an explicit measure of edge persistence across resize. For each train→test pair, we compute the activation frequency of every directed edge during evaluation and compare the resulting frequency matrices. The Pearson correlation between edge frequencies is consistently high (0.79–0.95), indicating that edges that are frequently used at the train population size tend to remain frequently used after resizing. Moreover, the Jaccard overlap of the top-$10\%$ most frequently used edges lies between 0.67 and 0.88 across all settings, showing that the majority of high-utility communication channels are reused rather than being replaced by entirely new links. Overlaps at $5\%$ are slightly lower due to the small set size and ranking noise, while overlaps at $20\%$ remain clearly above chance, suggesting that even a broader band of mid- to high-frequency edges is largely preserved.

### C.3.5 SYSTEM-LEVEL PROFILING

We further profile system-level costs as the number of agents increases on the Battle scenario (Tab. 8). SOPS maintains a small realized out-degree per agent across all scales, in between the ultra-sparse EC-O (one fixed neighbor) and the denser EC-S that always activates all exponential candidates (5–7 neighbors), and far below the fully connected GACG whose degree grows linearly with $N$. This sparse yet expressive communication comes with moderate overhead: SOPS runs slower and uses more memory than the non-communicating VDN and EC-O/EC-S baselines, but remains well within the same order of magnitude in both step/s and GPU memory. In contrast, GACG exhibits much lower throughput and rapidly increasing memory and activation footprints

as $N$ grows, reflecting its quadratic communication complexity. Overall, the profile confirms that SOPS achieves scalable communication with a small and slowly growing per-agent neighbor budget.

### C.3.6 VISUALIZATION FOR ZERO-SHOT TRANSFER

In Fig. 9, we visualize the dynamic communication subgraphs induced by SOPS for a representative focal agent during zero-shot transfer on Battle scenarios. Each subfigure corresponds to a specific time step (t = 20, 25, 30, 35, 40, 45), capturing the evolution of communication patterns. Red and blue squares denote the SOPS-controlled team and opponents, respectively, with numerical labels above each panel indicating the remaining agent counts. The arrows highlight the focal agent's dynamically evolving communication links, revealing how SOPS maintains structured interaction topologies across varying population sizes and time steps.

In Fig. 10 we visualize the spatial behavior of the learned SOPS policies under zero-shot transfer on the Battle scenarios. Each row corresponds to a different train→test population resize (from top to bottom: 20→42, 20→64, 42→100, 42→121 agents), and each column shows a snapshot at different rollout times (t = 0, 25, 50, 75). Red squares denote the SOPS-controlled team and blue squares the opponent; the numbers above each panel indicate the remaining agents on each side. At t = 0 the transferred policies reproduce the characteristic lattice-like initial formations learned during training, now instantiated at larger population sizes. As time evolves, the agents consistently form coherent battle fronts and concentrated engagement zones rather than degenerating into random or fragmented patterns, and the qualitative engagement strategy (front-line formation, gradual encirclement, and clean-up of remaining opponents) is preserved across all resize settings. These visualizations illustrate that SOPS-induced communication graphs support robust zero-shot transfer, yielding stable and interpretable large-scale behaviors even when the population size at test time differs substantially from training.

Table 8: System-level profiling with varying numbers of agents. We report wall-clock environment steps per second (step/s), peak GPU memory, and estimated activation size, together with the realized average out-degree as effective per-agent neighbors on the Battle scenario.

| Env ($N$) | Method | Avg. out-degree | step/s | GPU mem (GB) | Activation (GB) |
|---|---|---|---|---|---|
| Battle-20 | SOPS | 3.1 | 100.8 | 4.26 | 1.87 |
| Battle-20 | EC-S | 5.0 | 117.4 | 3.83 | 1.33 |
| Battle-20 | EC-O | 1.0 | 123.5 | 3.57 | 1.29 |
| Battle-20 | GACG | 19.0 | 34.0 | 8.54 | 3.55 |
| Battle-20 | VDN | 0.0 | 179.2 | 2.19 | 0.73 |
| Battle-42 | SOPS | 3.5 | 68.6 | 8.93 | 3.95 |
| Battle-42 | EC-S | 6.0 | 72.8 | 7.57 | 2.68 |
| Battle-42 | EC-O | 1.0 | 74.5 | 7.29 | 2.65 |
| Battle-42 | GACG | 41.0 | 30.9 | 14.52 | 5.80 |
| Battle-42 | VDN | 0.0 | 105.7 | 4.76 | 1.14 |
| Battle-64 | SOPS | 3.9 | 50.4 | 13.58 | 6.04 |
| Battle-64 | EC-S | 7.0 | 55.2 | 11.43 | 5.23 |
| Battle-64 | EC-O | 1.0 | 60.0 | 11.20 | 5.31 |
| Battle-64 | GACG | 64.0 | 27.6 | 20.44 | 9.51 |
| Battle-64 | VDN | 0.0 | 78.8 | 6.88 | 1.75 |
| Battle-100 | SOPS | 4.2 | 38.3 | 18.64 | 8.53 |
| Battle-100 | EC-S | 7.0 | 41.1 | 16.03 | 7.26 |
| Battle-100 | EC-O | 1.0 | 44.0 | 15.85 | 7.18 |
| Battle-100 | GACG | 100.0 | 23.1 | 33.81 | 14.24 |
| Battle-100 | VDN | 0.0 | 65.6 | 10.57 | 2.64 |
| Battle-121 | SOPS | 4.3 | 34.5 | 23.72 | 12.70 |
| Battle-121 | EC-S | 7.0 | 37.0 | 20.55 | 8.82 |
| Battle-121 | EC-O | 1.0 | 40.5 | 20.13 | 8.68 |
| Battle-121 | GACG | 121.0 | 14.5 | 48.50 | 20.49 |
| Battle-121 | VDN | 0.0 | 50.8 | 12.83 | 3.28 |

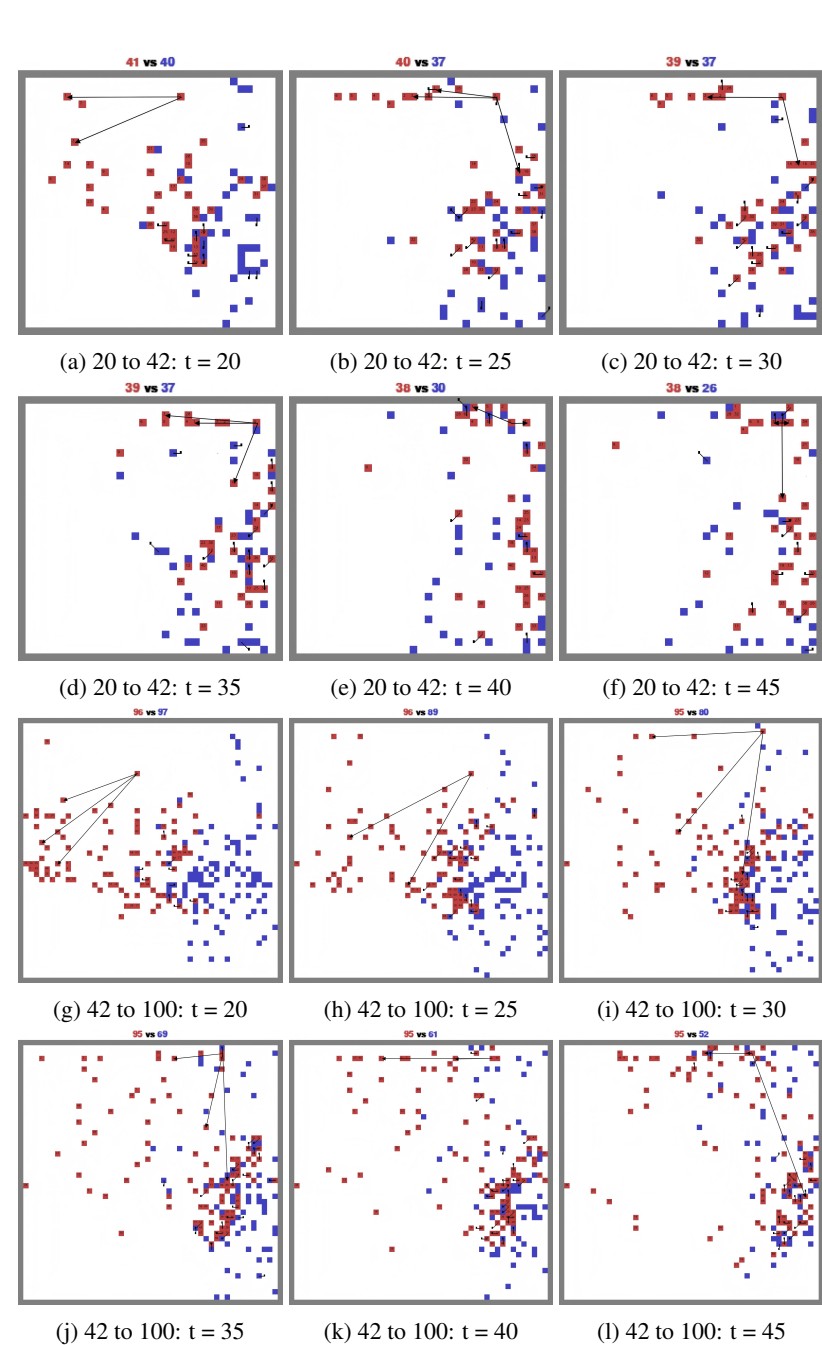

Figure 9: Visualization of SOPS-induced dynamic communication subgraphs under zero-shot transfer on Battle scenarios for a representative focal agent

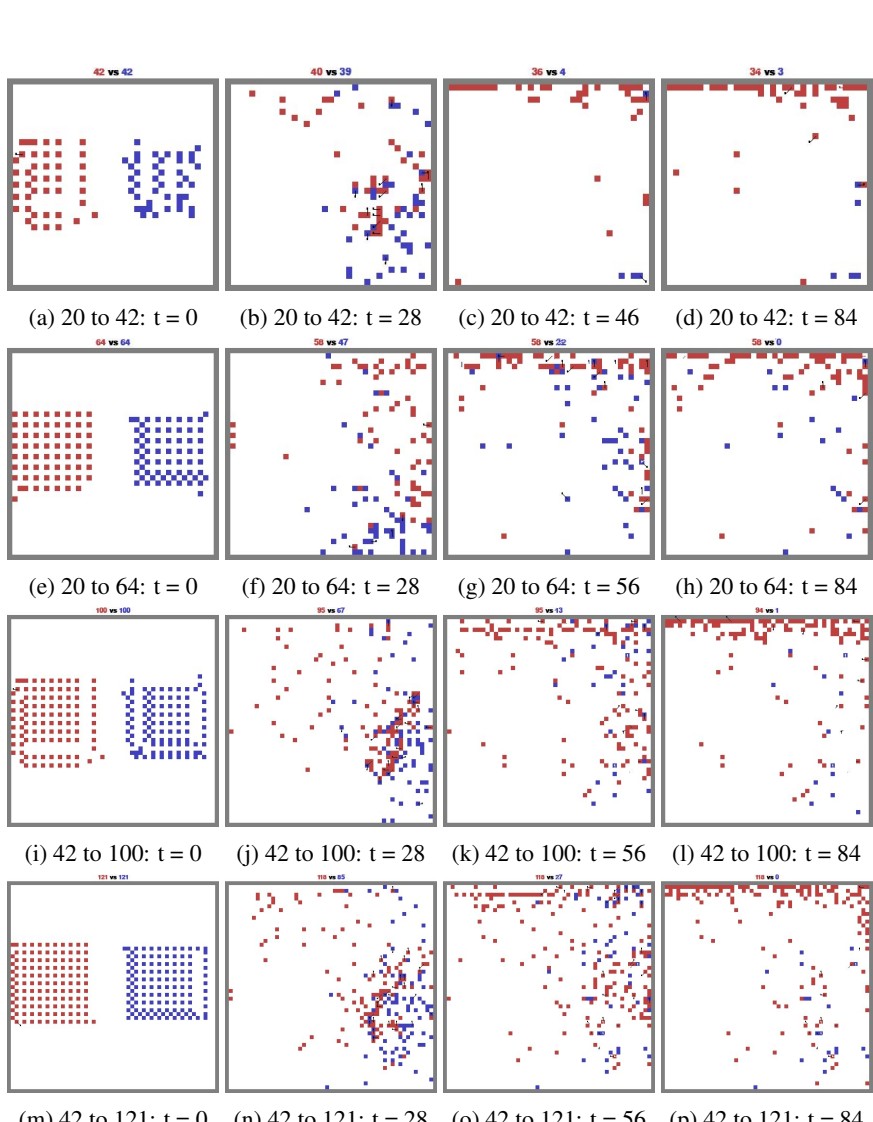

(a) 20 to 42: t = 0    (b) 20 to 42: t = 28    (c) 20 to 42: t = 46    (d) 20 to 42: t = 84

(e) 20 to 64: t = 0    (f) 20 to 64: t = 28    (g) 20 to 64: t = 56    (h) 20 to 64: t = 84

(i) 42 to 100: t = 0   (j) 42 to 100: t = 28   (k) 42 to 100: t = 56   (l) 42 to 100: t = 84

(m) 42 to 121: t = 0   (n) 42 to 121: t = 28   (o) 42 to 121: t = 56   (p) 42 to 121: t = 84

Figure 10: Visualization for zero-shot transfer performance on Battle scenarios.

