# OpenReview forum: "Sparse Topology Pairwise Scoring for Large-Scale Multi-Agent Reinforcement Learning"
_ICLR.cc/2026/Conference — Submitted to ICLR 2026_

### Official Review · Reviewer_tKLs · 2025-10-29

**Soundness:** 4
**Presentation:** 4
**Contribution:** 4
**Rating:** 6
**Confidence:** 4

**Summary:**

To address the issues of excessive communication overhead, poor scalability, and the difficulty of balancing complexity and adaptability in existing methods for large-scale cooperative multi-agent reinforcement learning (MARL) under the Centralized Training with Decentralized Execution (CTDE) paradigm, this study proposes a scalable communication mechanism called SOPS (Sparse Topology Pairwise Scoring). Its core design idea decouples "global reachability" from "task-adaptive selection": an exponential graph is used as the communication backbone to ensure efficient multi-hop information dissemination, leveraging its properties of small diameter and near-linear cost. On this basis, a lightweight pairwise scoring network dynamically generates task-adaptive sparse subgraphs by integrating agent states and edge-type embeddings. To solve the differentiability problem of discrete subgraph sampling, Gumbel-Sigmoid reparameterization with linear temperature annealing is introduced to enable end-to-end training. Meanwhile, auxiliary tasks (global state recovery or contrastive learning) are designed to ensure the task relevance of communication content. Additionally, this study verifies the compatibility of SOPS with mainstream value-based MARL algorithms and its zero-shot transfer capability across different agent scales, providing a new solution for the design of communication mechanisms in large-scale MARL.

**Strengths:**

(1) The study accurately captures the core pain points in large-scale MARL, such as communication bandwidth explosion and the difficulty of balancing topology flexibility and scalability. The proposed decoupling approach—"using a backbone network to ensure global reachability and pairwise scoring to achieve dynamic selection"—effectively breaks through the limitations of existing methods.
(2) Gumbel-Sigmoid reparameterization is introduced to solve the non-differentiability issue of discrete sampling, avoiding the high variance defect of traditional score-function methods. The linear temperature annealing strategy further optimizes the balance between exploration and exploitation, and the design of auxiliary tasks prevents meaningless communication.
(3) SOPS is designed as a "plug-and-play" module, compatible with mainstream value-based MARL algorithms such as IQL and QMIX. It executes fully in a decentralized manner without relying on a central proxy, making it easy to integrate into existing CTDE frameworks.
(4) Experiments cover multiple task types, including adversarial pursuit, team battle, and infrastructure management. They focus on verifying the method's performance in dimensions such as overall performance, convergence, and cross-scale transfer.

**Weaknesses:**

(1) The robustness of the exponential graph backbone in scenarios such as agent failure and dynamic topology disruption is not discussed, making it impossible to determine the method's applicability in unstable environments.
(2) All existing experiments are based on homogeneous agents (with consistent action spaces and roles). However, large-scale MARL scenarios in the real world often involve heterogeneous agents (with different capabilities and responsibilities).
(3) In the related work section, there is no in-depth comparison of SOPS with methods such as GTDE (Grouped Training with Decentralized Execution) and ExpoComm (fixed exponential topology) in terms of technical routes (e.g., grouping strategies vs. sparse subgraph selection) and applicable scenarios. The relative advantages and application boundaries of SOPS are not clearly highlighted, and the connection to and breakthroughs over existing achievements in the field are not sufficiently elaborated.
(4) The manuscript contains floating figures that are not explicitly referenced or explained in the main text. These figures lack contextualization—for instance, no descriptions of their purpose, the insights they convey, or how they support the core claims of the study.
(5) Some references in the manuscript are relatively old; it is recommended to replace them with more recent studies related to large-scale MARL.

**Questions:**

See the Weaknesses.

**Details Of Ethics Concerns:**

No Details Of Ethics Concerns.

---

> ### Author Response · Authors · 2025-11-22
>
> Thank you for your positive feedback! Regarding your insightful suggestions, we have updated the manuscripts accordingly and would like to provide detailed clarifications below. If you have any follow-up questions or comments, please let us know, and we will be happy to discuss further.
>
> **Q1:**
> > The robustness of the exponential graph backbone in scenarios such as agent failure and dynamic topology disruption is not discussed, making it impossible to determine the method's applicability in unstable environments.
>
> **A1:** Thank you for this important comment. **SOPS inherits robustness from its exponential graph backbone, which maintains connectivity and fast mixing under edge/node failures** [1, 2]. Crucially, its adaptive scoring and multi-hop aggregation do not rely on fixed edges—attention is dynamically redistributed over available neighbors. Although we do not run dedicated failure experiments, several of our existing results are consistent with robustness to topology perturbations: Fig.1 shows resilience under sparsification, and Fig.4 demonstrates stable performance under drastic topology changes (zero-shot transfer). Please see Appendix C.3.2 for a more detailed discussion.
>
> An interesting extension is to integrate SOPS with failure-aware routing or health signals from the system, allowing the scorer to downweight persistently unreliable neighbors. A thorough empirical study of such robustness mechanisms in real-world unstable environments is left to future work.
>
> **Q2:**
> > All existing experiments are based on homogeneous agents (with consistent action spaces and roles). However, large-scale MARL scenarios in the real world often involve heterogeneous agents (with different capabilities and responsibilities).
>
> **A2:** Thank you for your insightful comment! We agree that our current experimental suite focuses on environments with homogeneous agents, as is common in large-scale MARL benchmarks such as MAgent and IMP. Importantly, however, **SOPS does not rely on agent homogeneity at the algorithmic level**. The exponential backbone is defined over agent indices and is agnostic to their roles or capabilities, and the pairwise scoring module operates on learned embeddings. In heterogeneous settings, these embeddings can naturally incorporate agent-type or capability features, allowing the scorer to learn differentiated, role-aware connectivity patterns on top of the same sparse backbone.
>
> At the same time, we recognize that many real-world large-scale systems involve strongly heterogeneous agents where highly targeted, role-specific communication may be especially beneficial. Systematically evaluating SOPS in such strongly heterogeneous scenarios (and **extending the scorer with explicit type-aware edge features**) is a natural next step. We have clarified this scope in the paper and added a dedicated discussion in Appendix C.3.3, highlighting heterogeneity-aware extensions as a promising direction for future work while preserving the efficiency and scalability benefits of the exponential topology.
>
> **Q3:**
> > In the related work section, there is no in-depth comparison of SOPS with methods such as GTDE and ExpoComm (fixed exponential topology) in terms of technical routes (e.g., grouping strategies vs. sparse subgraph selection) and applicable scenarios. The relative advantages and application boundaries of SOPS are not clearly highlighted, and the connection to and breakthroughs over existing achievements in the field are not sufficiently elaborated.
>
> **A3:** Thank you for this insightful comment. We have added a targeted comparison of SOPS with GTDE and ExpoComm in the sub-section “Communication Learning and Topology Design” in Related Work, where we explicitly contrast their technical routes: GTDE relies on fixed groupings over dense graphs with unbounded bandwidth and population-dependent structure, while ExpoComm uses a fixed exponential topology that lacks adaptivity. In contrast, SOPS retains the exponential backbone only for reachability and learns a task-adaptive subgraph via differentiable pairwise scoring under strict bandwidth constraints.
>
> **Q4:**
> > The manuscript contains floating figures that are not explicitly referenced or explained in the main text. These figures lack contextualization—for instance, no descriptions of their purpose, the insights they convey, or how they support the core claims of the study.
>
> **A4:** Thank you for this important observation. We have carefully revised the manuscript to ensure that every figure and table is now explicitly referenced and contextualized in the main text, including both original and newly added visualizations. Each figure now includes a clear description of its purpose, key insights, and how it supports our claims.

---

> > ### Author Response · Authors · 2025-11-22
> >
> > **Q5:**
> > > Some references in the manuscript are relatively old; it is recommended to replace them with more recent studies related to large-scale MARL.
> >
> > **A5:** Thank you for this valuable feedback, and we have updated the Introduction section to include more recent, relevant studies on large-scale MARL [3–6], and have highlighted these additions in blue in the revised manuscript. Specifically, [3] investigates reinforcement learning in many-agent settings under partial observability, while [4] and [5] propose attention-based mean-field approximations for very large or homogeneous agent populations. In addition, [6] provides a recent survey of large-population systems and scalable MARL, helping to position our contribution within this emerging line of work.
> >
> > [1] Ying, Bicheng, et al. "Exponential graph is provably efficient for decentralized deep training." *Advances in Neural Information Processing Systems* 34 (2021)
> >
> > [2] Chen, Yiming, et al. "Accelerating gossip SGD with periodic global averaging." *International Conference on Machine Learning*. PMLR, 2021.
> >
> > [3] He, Keyang, Prashant Doshi, and Bikramjit Banerjee. "Reinforcement learning in many-agent settings under partial observability." *Uncertainty in Artificial Intelligence*. PMLR, 2022.
> >
> > [4] Hao, Qianyue, et al. "Gat-mf: Graph attention mean field for very large scale multi-agent reinforcement learning." *Proceedings of the 29th ACM SIGKDD Conference on Knowledge Discovery and Data Mining*. 2023.
> >
> > [5] Shike, Yang, Li Jingchen, and Shi Haobin. "Mix-attention approximation for homogeneous large-scale multi-agent reinforcement learning." *Neural Computing and Applications* 35.4 (2023)
> >
> > [6] Cui, Kai, et al. "A survey on large-population systems and scalable multi-agent reinforcement learning." *arXiv preprint arXiv:2209.03859*

---

> > > ### Comment · Reviewer_tKLs · 2025-11-24
> > >
> > > I have carefully reviewed the revised manuscript and the authors’ response to my initial comments. Overall, the authors have addressed all the concerns raised in the first review comprehensively and effectively.
> > > Thanks for the author's explanation. I will maintain my positive rating.

---

> > > > ### Author Response · Authors · 2025-11-24
> > > >
> > > > We sincerely appreciate the reviewer for carefully re-evaluating our revised manuscript and for the positive follow-up comment. We are glad that the additional experiments, analyses, and explanations helped to address the earlier concerns and improve the clarity and quality of the paper. If you feel that the revision now better reflects the contribution of this work, we will be very grateful if you would consider a raise in the final rating.

---

> > ### Comment · Reviewer_tKLs · 2025-11-24
> >
> > I have carefully reviewed the revised manuscript and the authors’ response to my initial comments. Overall, the authors have addressed all the concerns raised in the first review comprehensively and effectively.
> > Thanks for the author's explanation. I will maintain my positive rating.

---

### Official Review · Reviewer_tRJ9 · 2025-10-31

**Soundness:** 3
**Presentation:** 3
**Contribution:** 3
**Rating:** 6
**Confidence:** 3

**Summary:**

This paper introduces SOPS, a scalable communication mechanism for large-scale cooperative MARL. It addresses the challenge of communication overhead in large agent populations by combining a fixed exponential backbone topology with a learned, sparse, and task-adaptive subgraph. The subgraph is generated using a pairwise scoring network and Gumbel-Sigmoid reparameterization, enabling end-to-end differentiable training. SOPS is designed to be plug-and-play with common CTDE-based learners like IQL and QMIX. Experiments on IMP benchmarks show that SOPS achieves higher returns, faster convergence, and robust zero-shot transfer across varying agent population sizes. Ablation studies further validate the effectiveness of the Gumbel temperature annealing and reparameterization strategy.

**Strengths:**

One of the key strengths of the paper is its scalability. By leveraging the exponential graph structure, the communication cost grows near-linearly with the number of agents, while maintaining a small diameter for rapid information dissemination. The learned sparse subgraph adds adaptability, allowing the communication structure to evolve with task dynamics. The use of Gumbel reparameterization ensures stable and efficient training of discrete communication links, which is often a challenge in MARL.

The empirical results across benchmarks such as MAgent and IMP demonstrate superior performance in terms of convergence speed, final returns, and zero-shot transfer to larger agent populations. Ablation studies further validate the importance of temperature annealing and reparameterization in achieving robust learning.

**Weaknesses:**

Although the experimental results consistently outperform other models, the statistical significance of these results appears limited. This raises the question of whether it is safe to draw strong conclusions based solely on such outcomes. Without rigorous statistical validation, the observed performance gains may not be robust or generalizable.

One of the stated motivations or contributions of the paper is to ensure that the underlying communication backbone follows a power-law structure to enable efficient information dissemination. However, it remains unclear how the subgraph sampling strategy guarantees that the final communication graph retains the properties of a power-law network. Further clarification or empirical evidence would strengthen this claim.

**Questions:**

Please see the above two comments.

---

> ### Author Response · Authors · 2025-11-22
>
> Thank you for your positive feedback! Regarding your questions and suggestions,  we provide detailed clarifications below. If you have any follow-up questions or comments, please let us know, and we will be happy to discuss further.
>
> **Q1:**
> > Although the experimental results consistently outperform other models, the statistical significance of these results appears limited. This raises the question of whether it is safe to draw strong conclusions based solely on such outcomes. Without rigorous statistical validation, the observed performance gains may not be robust or generalizable.
>
> **A1:** We sincerely thank the reviewer for raising the important point regarding statistical significance. For MAgent, we agree that in some smaller-scale settings, the 95% confidence bands of SOPS and EC-O partially overlap, so the gains may not always appear strikingly large at first glance. This is expected: EC-O is already a very strong baseline in these regimes, and both methods approach the performance ceiling (e.g., win rate close to 1), which naturally compresses and overlaps the confidence ranges. Even in these cases, SOPS is at least competitive and often converges slightly faster. The more important trend, however, emerges as the population size increases. On larger MAgent maps, SOPS consistently converges faster, reaches higher plateaus, and exhibits more stable late-training behavior than EC-O and EC-S. To quantify this effect more systematically, we have additionally computed area-under-the-curve (AUC) metrics for the evaluation curves on AdversarialPursuit and Battle (see new Tabs. 4 and 5 in the revised manuscript), and we further summarize two-sample t-test p-values on AUC for large-scale scenario in the table below. On the larger maps (APurs101, Battle64/100/121), two-sample t-tests on AUC across three seeds yield p-values below 0.05 when comparing SOPS against both EC-O and EC-S.
>
> |Compare with | APurs101   | Battle64 | Battle100 | Battle121 |
> |-------|---------|----------|-----------|-----------|
> | EC-O  | 0.0211  | 0.0235   | 0.0301    | 0.0066    |
> | EC-S  | 0.0079  | 0.0017   | 0.0053    | 0.0016    |
>
> Regarding the IMP benchmark, our intention is not to claim large, statistically overwhelming gains, but rather to provide complementary evidence of robustness on a very different engineering domain. IMP is designed around strong expert-based heuristic baselines and cost-oriented objectives, and prior work already reports relatively limited margins between advanced MARL methods and these heuristics in several campaign-cost configurations [1]. From the results in Tab. 1, we empirically observe that in every IMP environment, SOPS remains competitive with the best existing CTDE-style learner and often achieves a small but consistent lead. Moreover, to demonstrate that these gains are not tied to a single backbone, we newly extended SOPS from its original QMIX-based implementation to two additional value-decomposition methods, QPLEX and SHAQ, under the same training setup and hyperparameters. The results, reported in new Tab. 2, show that attaching SOPS to each learner consistently improves performance over its backbone, and the best or second-best method in every scenario is always a “+SOPS’’ variant. Accordingly, we interpret the IMP results as evidence that SOPS can be integrated into a realistic infrastructure-management benchmark without harming performance and often providing modest improvements over strong CTDE baselines.

---

> > ### Author Response · Authors · 2025-11-22
> >
> > **Q2:**
> > > One of the stated motivations or contributions of the paper is to ensure that the underlying communication backbone follows a power-law structure to enable efficient information dissemination. However, it remains unclear how the subgraph sampling strategy guarantees that the final communication graph retains the properties of a power-law network. Further clarification or empirical evidence would strengthen this claim.
> >
> > **A2:** We appreciate the reviewer’s request for clarification. Our intention is not to enforce a strict power-law degree distribution on the final communication graph. Instead, we **rely on a static exponential candidate skeleton** that provides a logarithmic-diameter, small-world–like backbone, where each agent considers only a small number of candidates (growing at most logarithmically with the population size), leading to a sparse graph even at large scales. **The goal of the subgraph sampler is therefore to preserve the desirable properties of this backbone (sparsity and fast multi-hop reachability), rather than to match an idealized power-law network in the strict sense**.
> >
> > To substantiate that the learned subgraph indeed retains these properties after resizing, we have added the new subsection Appendix C.3.4 **Learned-Graph Diagnostics**. Tab. 6 shows that, across all train$\rightarrow$test populations in Fig. 4, the average out-degree stays within 3–4.6 edges per agent, **confirming that SOPS maintains a sparse and size-stable communication load**. At the same time, coverage within 3–4 hops remains high, indicating that the effective **diameter of the learned graph stays small and information can be propagated in a few communication rounds**. Tab. 7 further reports edge-frequency persistence between train and zero-shot populations: edge activation frequencies exhibit high Pearson correlation (0.79–0.95), and the top-10% most frequently used edges achieve substantial Jaccard overlap (0.67–0.88) across sizes. These diagnostics together show that the subgraph sampling strategy **preserves the scale-consistent communication patterns** induced by the exponential topology, which explains the strong zero-shot transfer behavior.
> >
> > [1] Leroy, Pascal, et al. "IMP-MARL: a suite of environments for large-scale infrastructure management planning via MARL." *Advances in neural information processing systems* 36 (2023)

---

### Official Review · Reviewer_eN33 · 2025-11-01

**Soundness:** 4
**Presentation:** 4
**Contribution:** 1
**Rating:** 4
**Confidence:** 5

**Summary:**

This paper proposes SOPS, a scalable communication strategy for large-scale MARL. The method builds upon a fixed exponential graph backbone to ensure efficiency, introducing a lightweight pairwise scoring module to dynamically learn a sparse, task-adaptive communication subgraph. By leveraging Gumbel-Sigmoid reparameterization for end-to-end training, SOPS aims to provide a plug-and-play solution that demonstrates strong performance and robust zero-shot transfer capabilities in multiple experiments.

**Strengths:**

1. The paper addresses the highly relevant and practical problem of scalable communication in MARL. The focus on balancing performance with communication cost is well-motivated and crucial for real-world applications.
2. The key innovation of this method seems to be the added pairwise-scoring module on top of the established exponential topology. This is an intuitive and sensible approach, as it introduces necessary flexibility for task-adaptive communication while retaining the guaranteed reachability of the backbone topology. The empirical results help validate the effectiveness of this design choice.

**Weaknesses:**

1. The method is presented as a refinement of the existing exponential topology paradigm. While the addition of the pairwise scoring module is a reasonable next step, the overall contribution seems like as a marginal improvement rather than a fundamentally new approach to scalable communication.
2. The empirical performance gains, while consistent, do not always appear to be statistically significant (standard deviation error bars often overlaps with the performance gain).


Minor:
Please number all equations for easier references.

**Questions:**

1. The computation of the pairwise score (line 252) appears to require embeddings from all candidate neighbors on the backbone. How are these embeddings shared before the communication links for the current timestep are established? Does this imply a pre-communication phase where agents broadcast their embeddings to their backbone neighbors, and if so, how is the overhead of this phase accounted for in the overall efficiency analysis?
2. Is the $l_{ij}$ computed at the receiver or sender side?
3. Is the hyperparameter $\tau$ consistent across all experiments, or are they tuned per environment? If they require tuning, could the authors provide some intuition or a methodology for selecting appropriate values, as this seems crucial for the method's stability and performance?
4. Could the authors provide some visualizations of the resulting dynamic subgraphs? Those could offer valuable intuition into the adaptive strategies that emerge and help readers better understand how SOPS contributes to the improved joint policies.

---

> ### Author Response · Authors · 2025-11-22
>
> Thank you for your constructive feedback. Following your minor, we have added equation numbers to all formulas for easier reference. Regarding your questions and suggestions,  we provide detailed clarifications below. If you have any follow-up questions or comments, please let us know, and we will be happy to discuss further.
>
> **W1:**
> > The method is presented as a refinement of the existing exponential topology paradigm. While the addition of the pairwise scoring module is a reasonable next step, the overall contribution seems like a marginal improvement rather than a fundamentally new approach to scalable communication.
>
> **A1:** We appreciate the reviewer’s careful assessment. Our method is indeed built on the static exponential topology paradigm introduced by ExpoComm, and we do not claim to “invent a new topology” from scratch. Instead, SOPS should be viewed as a concrete step towards addressing a structural limitation in current designs: **the lack of a mechanism that can interpolate between a global topology perspective and a local (pairwise) perspective in many-agent communication**. A purely local, task-oriented pairwise view can be very effective in small systems, but it quickly runs into scalability issues because the number of potential communication links grows as $O(N^2)$. We extract a time-varying subgraph from the static exponential graph. In this sense, SOPS does not discard the global perspective, but rather blends global and local views into a single topology-aware sparse routing mechanism. From an algorithmic and systems perspective, the pairwise scoring module is also not a trivial add-on. SOPS restricts scoring to the backbone edges and makes these discrete link decisions trainable end-to-end through a Gumbel-sigmoid reparameterization with linear temperature annealing.
>
> Empirically, this design yields consistent gains over static exponential baselines across MAgent and IMP, with the largest improvements appearing in the most challenging large-scale settings. In addition, we newly validate in Tab. 2 that SOPS is plug-and-play with multiple value-decomposition learners, where attaching SOPS consistently improves over each backbone. Appendix C.3.4 provides zero-shot graph diagnostics: SOPS retains the exponential backbone’s coverage while adapting to new population sizes, confirming it's  effectiveness and robustness.
>
> **W2:**
> > The empirical performance gains, while consistent, do not always appear to be statistically significant (standard deviation error bars often overlap with the performance gain).
>
> **A2:** We sincerely thank the reviewer for raising the important point regarding statistical significance. For MAgent, we agree that in some smaller-scale settings, the 95% confidence bands of SOPS and EC-O partially overlap. This is expected: EC-O is already a very strong baseline in these regimes, and **both methods approach the performance ceiling** (e.g., win rate close to 1). Even in these cases, SOPS is at least competitive and often converges slightly faster. However, **on larger MAgent maps, SOPS consistently converges faster, reaches higher plateaus, and exhibits more stable late-training behavior** than EC-O and EC-S. We have additionally computed area-under-the-curve (AUC) metrics for the evaluation curves on AdversarialPursuit and Battle (see new Tabs. 4 and 5 in the revised manuscript), and we further summarize two-sample t-test p-values on AUC for large-scale scenario in the table below, all of which are statistically significant (p < 0.05).
>
> | Compare with | APurs101 | Battle64 | Battle100 | Battle121 |
> | --- | --- | --- | --- | --- |
> | EC-O | 0.0211 | 0.0235 | 0.0301 | 0.0066 |
> | EC-S | 0.0079 | 0.0017 | 0.0053 | 0.0016 |
>
> Regarding the IMP benchmark, our intention is not to claim large, statistically overwhelming gains, but rather to **provide complementary evidence of robustness on a very different engineering domain**. IMP is designed around strong expert-based heuristic baselines and cost-oriented objectives, and prior work already reports relatively limited margins between advanced MARL methods and these heuristics in several campaign-cost configurations [1]. We newly extended SOPS from its original QMIX-based implementation to two additional methods under the same training setup and hyperparameters. The results reported in new Tab. 2 show that **attaching SOPS to each learner consistently improves performance over its backbone, and the best or second-best method in every scenario is always a “+SOPS’’ variant**. Accordingly, we interpret the IMP results as evidence that SOPS can be integrated into a realistic infrastructure-management benchmark without harming performance and often providing modest improvements over strong CTDE baselines.
>
> [1] Leroy, Pascal, et al. "IMP-MARL: a suite of environments for large-scale infrastructure management planning via MARL." *Advances in neural information processing systems* 36 (2023)

---

> > ### Author Response · Authors · 2025-11-22
> >
> > **Q1:**
> > > The computation of the pairwise score (line 252) appears to require embeddings from all candidate neighbors on the backbone. How are these embeddings shared before the communication links for the current timestep are established? Does this imply a pre-communication phase where agents broadcast their embeddings to their backbone neighbors, and if so, how is the overhead of this phase accounted for in the overall efficiency analysis?
> >
> > **A3:** The pairwise scorer indeed takes as input the embeddings of candidate neighbors on the backbone, but this **does not require an extra pre-communication round**. Under the standard CTDE setup, all agent embeddings for the current timestep are **computed on the centralized controller**, so the scorer has direct access to the full tensor without any agent-to-agent exchange. The only communication counted is the sparse message passing along sampled edges during execution; no additional pre-communication phase exists, and the asymptotic communication cost remains unchanged.
> >
> > If a fully decentralized implementation is desired, agents can instead reuse the neighbor embeddings received in the previous timestep as inputs to the scorer. This variant likewise introduces no extra broadcast and does not change the asymptotic communication cost.
> >
> > **Q2:**
> > > Is the $l_{ij}$ computed at the receiver or sender side?
> >
> > **A4:** $l_{ij}$ is conceptually computed on the sender side. In our implementation, it is computed centrally on the controller under the CTDE setup, but it is a sender-anchored score for the directed candidate edge $i \to j$: the scorer evaluates all backbone candidates of each sender $i$ in a single forward pass on the learner. In the revised manuscript, we clarify this design in Sec. 4.1 by explicitly stating that “$\ell^t_{ij}$ is a sender-anchored score for the directed edge $i \to j$; under CTDE, these scores are computed centrally on the learner from the full embedding tensor, but they are attached to outgoing edges from each sender.
> >
> > Similarly, if a fully decentralized variant is desired, senders can instead compute $l_{ij}$ locally using the cached neighbor embeddings from the previous timestep. Compared to a receiver-side pull design (which typically requires a request/ack control signal), the sender-side variant needs no extra broadcast or handshake, and aligns with our sender-anchored scoring.
> >
> > **Q3:**
> > > Is the hyperparameter $\tau$ consistent across all experiments, or are they tuned per environment? If they require tuning, could the authors provide some intuition or a methodology for selecting appropriate values, as this seems crucial for the method's stability and performance?
> >
> > **A5:** We thank the reviewer for raising this point. In all our reported experiments, we **use a single linearly annealed schedule**, shared across all environments and agent scales, without per-environment tuning. According to prior work, smaller $\tau$ drive samples closer to one-hot discrete draws but increase gradient variance, whereas larger $\tau$ produce smoother, more uniform samples with lower-variance yet weaker gradients [2]. Motivated by this trade-off, we employ a simple linear schedule that starts from a relatively high temperature (1.0) for stable early optimization and anneals to a moderate value (0.3) to obtain a more stable topology later. Our ablation study on the temperature schedule (Fig. 5(a)) shows that **this linear annealing consistently outperforms both fixed temperatures and exponential annealing**. And we observe similar trends across different configurations. These observations indicate that this schedule is a robust and reliable default across both MAgent and IMP, even without environment-specific tuning.
> >
> > For practitioners applying SOPS to new domains, we suggest **treating $\tau$ as a standard optimization hyperparameter**. A coarse sweep over $(\tau_\text{max}, \tau_\text{min})$ is often sufficient: if training is slow and gates remain near 0.5, faster annealing or a slightly smaller $\tau_\text{max}$ can help; if training is unstable and the topology changes too abruptly, a larger $\tau_\text{min}$ mitigates overly hard gates. In our experience, **once a stable pair $(\tau_\text{max}, \tau_\text{min})$ is identified on a representative scenario, the same values typically generalize well without further per-environment adjustments**.
> >
> > [2] Jang, Eric, Shixiang Gu, and Ben Poole. "Categorical Reparameterization with Gumbel-Softmax." *International Conference on Learning Representations*. 2017.

---

> > > ### Author Response · Authors · 2025-11-22
> > >
> > > **Q4:**
> > > > Could the authors provide some visualizations of the resulting dynamic subgraphs? Those could offer valuable intuition into the adaptive strategies that emerge and help readers better understand how SOPS contributes to the improved joint policies.
> > >
> > > **A6:** We appreciate this suggestion. To provide a more intuitive view of the learned communication patterns, we have added a new visualization section in the appendix. In the Fig. 9, we overlay the SOPS-sampled communication edges on the spatial Battle map for zero-shot resize settings (20→42 and 42→100 agents) at several rollout times. Red and blue squares denote the two teams, and the black arrow line segments show the active SOPS edges for a single focal SOPS agent (agent 0) at each timestep. And in Fig. 10 we visualize the spatial behavior of the learned SOPS policies under zero-shot transfer on more Battle scenarios.
> > >
> > > In addition, Appendix C.3.4 provides quantitative diagnostics of the learned sparse graphs under population resizing. Tab. 6 shows that across all train→test pairs, the realized out-degree remains in a narrow range around 3–4.5, substantially smaller than the exponential candidate set, indicating that SOPS consistently maintains a low per-agent communication load even after resizing. At the same time, coverage within 3–4 hops quickly approaches 0.9 or higher in moderate resize regimes (e.g., Battle 42→81, AdversarialPursuit 25→45), and remains well above what would be expected from random sparse graphs even for more extreme enlargements (e.g., Battle 20→100/121). Tab. 7 further shows that the Pearson correlation of edge activation frequencies stays high (0.79–0.95) and that the Jaccard overlap of top-10% most frequently used edges is substantial across settings, meaning that high-utility communication channels are largely preserved rather than being replaced by entirely new links. Together, these findings characterize the adaptive strategy induced by SOPS: it preserves a small, stable set of high-value edges while adjusting which local and long-range neighbors are emphasized at different scales, enabling efficient global information flow and contributing to the improved joint policies observed in our experiments.

---

### Official Review · Reviewer_bpJA · 2025-11-02

**Soundness:** 2
**Presentation:** 2
**Contribution:** 2
**Rating:** 4
**Confidence:** 4

**Summary:**

The paper proposes Sparse tOpology Pairwise Scoring (SOPS), a communication scheme for large-scale cooperative MARL. SOPS fixes an exponential backboneand learns a task-adaptive sparse subgraph on top of it via a lightweight pairwise scoring network and Gumbel-Sigmoid sampling with linear temperature annealing. Messages are aggregated with cross-attention and the policy is trained under CTDE, with auxiliary objectives for message grounding. Experiments on MAgent and IMP benchmarks report higher returns/win-rates and faster convergence than baselines, plus zero-shot transfer from small to larger populations. Ablations favor linear annealing and reparameterized gradients over SFE/ARM.

**Strengths:**

1. The exponential backbone keeps degree $O(logN)$ and diameter $O(log N)$), a sensible starting point for large populations. The paper illustrates broadcast coverage advantages over ER/Torus variants.

2. The pairwise scorer and Gumbel-Sigmoid yield a trainable discrete subgraph with a practical linear annealing schedule. Ablations support this choice over SFE/ARM or fixed temperatures.

3. Across six MAgent settings and multiple IMP scenarios, SOPS typically learns faster and reaches better asymptotes than VDN, GACG, and ExpoComm variants.

4. Demonstrations of train to test scaling, e.g., 20 to 64/81/100/121 agents, 42 to 81/100/121, are valuable for large-population MARL.

**Weaknesses:**

1. The paper asserts near-linear communication/memory, yet provides no wall-clock, GPU-memory, or activation-size profiling vs N, nor bandwidth per step or per agent.

2. Strong attention-based or topology-learning methods beyond the chosen set are absent. In addition, GACG is excluded from zero-shot transfer on architectural grounds, weakening the generalization claim.

3. Since SOPS is presented as “plug-and-play” with common CTDE learners, it is surprising that results are only with QMIX. Most up-to-date CTDE-sytle methods are not shown.

4. The method samples hard edges via a straight-through estimator but provides no diagnostics on gradient bias/variance, calibration of selection probabilities, learned degree/coverage distributions, or stability across seeds. Zero-shot transfer is only shown for Battle, and there is no analysis of which edges persist or adapt when resizing.

**Questions:**

1. Beyond EC-S/EC-O, please add ablations that hold the backbone but remove/alter the scorer (e.g., random/top-k by static features) and vary the edge budget K to separate gains from topology vs learned selection.

2. Could you (i) evaluate zero-shot transfer on IMP, not just Battle, and (ii) provide learned-graph diagnostics e.g., degree, hop-coverage over time, edge persistence across resize, to explain why SOPS transfers better?

3. Since SOPS is “plug-and-play,” please include more up-to-date CTDE-style baselines and discuss whether benefits persist under different value-decomposition learners.

4. Can you report wall-clock step/s, GPU memory, and activation sizes with number of agents and per-agent neighbors, for SOPS and each baseline, plus ablate message dimensionality? A figure showing reward vs throughput/memory would substantiate scalability claims.

---

> ### Author Response · Authors · 2025-11-22
>
> Thank you for your constructive feedback. Regarding your questions and suggestions,  we provide detailed clarifications below. If you have any follow-up questions or comments, please let us know, and we will be happy to discuss further.
>
> **Q1:** (1) Strong attention-based or topology-learning methods beyond the chosen set are absent. (2) In addition, GACG is excluded from zero-shot transfer on architectural grounds, weakening the generalization claim.
>
> **A1:** (1) We clarify that our selected baseline methods are already strong attention/topology-learning methods that are scalable to 40–100+ agents: ExpoComm’s EC-O/EC-S perform attention and topology selection on the exponential backbone, and GACG uses a fully-connected, group-aware attention graph over all agent pairs. Many other high-quality topology-learning or graph-attention methods rely on dense $O(N^2)$ message passing/GNN layers that are not practically runnable at our scales, which motivated our current baseline choices.
>
> (2) GACG is excluded from the zero-shot transfer experiments because its fully-connected coordination graph and group structure are tied to a fixed agent set and are not directly reusable when the number or identities of agents change, so applying a trained GACG to a different team size would require architectural changes or retraining rather than genuine zero-shot reuse. The relevant text in the revised manuscript has been updated accordingly and highlighted in blue.
>
> **Q2:** Add ablations to separate gains from topology vs learned selection.
>
> **A2:** Following your suggestion, we removed the scorer and performed an ablation on the edge budget K by selecting edges using Euclidean distance as the static feature. The results are shown in Fig. 5, subfigure (b). As shown in the figure, SOPS exhibits rapid improvement after the warm-up phase and converges stably to a high win rate. In contrast, the static baselines with K=2 and K=3 exhibit pronounced late-stage oscillations, indicating insufficient connectivity for stable coordination. The K=4 variant is more stable but learns slowly and fails to match SOPS’s final performance. These results indicate that while the exponential backbone already provides a strong inductive bias, the main additional gains in both final performance and stability come from the learned, adaptive neighbor selection.
>
> **Q3:** Inadequate evaluation of zero-shot transferability and learned graph structure: (i) Evaluate zero-shot transfer on IMP, not just Battle. (ii) Provide learned-graph diagnostics.
>
> **A3:** (i) Zero-shot transfer across different agent counts **is not applicable to the IMP benchmark due to the way the environment and observations are defined** in the original IMP paper. In IMP, the number of components/agents n_comp is a structural property of the MDP, and many observation fields scale linearly with n_comp — for example, obs_multiple, obs_all_d_rate, and the global state representation have dimensions proportional to n_comp · 30 [1]. Changing the environment from 50 to 100 components, therefore, alters the dimensionality of both the observation vector and the output space, making policies trained in the 50-agent scenario structurally incompatible with 100-agent scenario checkpoints. To better address your concern, we added Figs. 4(h) to 4(k) to show the zero-shot transfer performance of SOPS in the AdversarialPursuit scenario, where it remains consistently strong.
>
> [1] Leroy, Pascal, et al. "IMP-MARL: a suite of environments for large-scale infrastructure management planning via MARL." *Advances in neural information processing systems* 36 (2023)
>
> (ii) We thank the reviewer for this suggestion. We have added section **Learned-Graph Diagnostics** with test-time analyses of the learned communication graph across zero-shot settings (please refer to Appendix C.3.4).  First, we report the average out-degree and find that SOPS consistently maintains a sparse graph (3–4.5 outgoing edges per agent), close to the exponential candidate budget, and stable after resizing. Second, we compute multi-hop coverage and observe that, even under population enlargement, SOPS achieves high coverage within 3–4 hops, indicating a small effective diameter and fast information propagation. Third, we measure edge persistence by comparing test-time edge activation frequencies between train and zero-shot populations: Pearson correlations are high (0.8–0.95), and the top-10% most frequently used edges show substantial Jaccard overlap across sizes. These diagnostics together indicate that SOPS learns scale-consistent, sparse, and quickly mixing communication patterns that are largely preserved under resizing, which explains its strong zero-shot transfer. Full numerical results are provided in Tables 6 and 7 in Appendix C.3.4.

---

> > ### Author Response · Authors · 2025-11-22
> >
> > **Q4:**
> > > Since SOPS is “plug-and-play,” please include more up-to-date CTDE-style baselines and discuss whether benefits persist under different value-decomposition learners.
> >
> > **A4:** We sincerely thank the reviewer for this suggestion. Here, “plug-and-play’’ means that SOPS can be added as a modular communication layer to standard CTDE value-decomposition learners while keeping their architectures, losses, and hyperparameters unchanged, requiring only a thin wrapper to route agent embeddings through the SOPS scorer. To verify that SOPS is truly “plug-and-play’’ across CTDE-style learners, we additionally evaluated it on the IMP benchmark with two more recent value-decomposition methods, QPLEX and SHAQ, using the same training setup and hyperparameters. The results are summarized in the new Table 2. Across all three IMP scenarios, attaching SOPS to each learner consistently improves performance over its backbone, and the best or second-best method in every scenario is always a “+ SOPS’’ variant. This indicates that SOPS brings complementary benefits that persist under different and more expressive value-decomposition learners, rather than being tied to a particular CTDE baseline.
> >
> > **Q5:** Report the efficiency metrics for SOPS and each baseline, along with the number of agents and per-agent neighbors, and ablate the message dimensionality.
> >
> > **A5:** Thank you for the suggestion. In the revised manuscript, we add a new appendix section C.3.5 System-level profiling, where we systematically report wall-clock environment steps per second, peak GPU memory, and estimated activation size as functions of the number of agents and the realized average out-degree (effective per-agent neighbors), for SOPS and all baselines. We profile the Battle scenario at N = (20, 42, 64, 100, 121) under identical training configurations on a single GPU. For brevity, we show below the peak GPU memory (GB) versus the number of agents; the full results, including step/s and activation sizes, are provided in Appendix C.3.5. We observe that SOPS scales approximately linearly in memory with the number of agents and remains close to the exponential-topology baselines (EC-O/EC-S), while dense attention-based GACG exhibits substantially higher memory usage. VDN provides a communication-free lower bound.
> >
> > | Method | 20    | 42    | 64    | 100   | 121   |
> > |--------|-------|-------|-------|-------|-------|
> > | SOPS| 4.26  | 8.93  | 13.58 | 18.64 | 23.72 |
> > | EC-S| 3.83  | 7.57  | 11.43 | 16.03 | 20.55 |
> > | EC-O| 3.57  | 7.29  | 11.20 | 15.85 | 20.13 |
> > | GACG| 8.54  | 14.52 | 20.44 | 33.81 | 48.50 |
> > | VDN | 2.19  | 4.76  | 6.88  | 10.57 | 12.83 |
> >
> > For the message dimensionality, we clarify that the communication message for SOPS in our implementation is simply the per-agent hidden state of the underlying value-based learner; SOPS does not introduce an additional message embedding separate from the agent representation. Consequently, we use the same hidden dimension $d$ for all methods, following the standard configurations in prior work, so that differences in performance and scalability stem from the communication topology rather than representational capacity. Varying $d$ would scale compute and memory almost uniformly across all methods and is orthogonal to our main focus, so we keep $d$ fixed and instead provide the detailed system-level profiling in Appendix C.3.5.

---

### Comment · Area_Chair_b9Vc · 2025-11-24

Dear Reviewer,

Thanks for taking the time to review this work. The authors have responded to your reviews. Can you please have a look at the rebuttal and discuss with the authors?

Best Regards,

AC

---

### Author Response · Authors · 2025-12-01
**General Response**

Dear Reviewers, AC, SAC, and PC,

We extend our sincere gratitude for the time and effort you have dedicated to reviewing our manuscript.

In this paper, we propose SOPS, a new approach to addressing the scalability challenges of MARL in large-scale scenarios. We are grateful for reviewers' recognition of the soundness and the scalability of the approach. **We have comprehensively addressed all concerns of all reviewers.** All the added content in the revised manuscript has been highlighted **in blue**. Below, we concisely summarize how we addressed all major concerns and the corresponding revisions made to the manuscript. For clarity, we refer to Reviewer bpJA as R1, Reviewer eN33 as R2, Reviewer tRJ9 as R3, and Reviewer tKLs as R4.

---

**Key clarifications:**

- **Communication cost and implementation details (R2, Q1-Q2; R1, Q5):** All agent embeddings are computed once on the centralized learner under CTDE; sender-anchored scores $\ell_{ij}$ are obtained from this tensor, so **no extra pre-communication phase is required** and the asymptotic communication cost is unchanged. Messages are simply the per-agent hidden states with the same dimension $d$ as in the baselines.

- **Temperature schedule (R2, Q3):** We use a **single linearly annealed schedule** shared across all environments and agent counts, without per-environment tuning.

- **Correction of the “power-law” misunderstanding (R3, Q2):** We clarify that we **do not enforce a power-law degree distribution**. SOPS relies on a static exponential candidate skeleton to provide sparse, small-diameter connectivity, and the learned subgraph is designed to preserve these properties under a fixed bandwidth budget.

- **Robustness and heterogeneity (R4, Q1-Q2):** We strengthen our discussion by explaining **how SOPS inherits robustness properties** of exponential graphs under edge/node failures and **how adaptive scoring redistributes attention** without relying on fixed links (Appendix C.3.2). We also clarify that **SOPS does not assume homogeneous agents** and highlight how heterogeneity-aware extensions naturally arise through type-encoded embeddings (Appendix C.3.3).

---

**Added Experiments and Analysis:**

- **Learned-graph diagnostics (R1, Q3; R3, Q2):** Appendix C.3.4 reports realized out-degree, multi-hop coverage, edge-frequency correlations and Jaccard overlap for top-10% edges across $train \to test$ sizes. These results show that **SOPS learns sparse, small-diameter, and scale-consistent communication patterns** that persist under resizing.

- **Supplementary plug-and-play experiments (R1, Q4):** To further validate the plug-and-play nature of SOPS, we extend it from the original QMIX backbone to QPLEX and SHAQ on IMP under the same training setup (Tab. 2). In all scenarios, attaching SOPS consistently improves over each backbone, and **the best or second-best method is always a “+SOPS” variant**, supporting the plug-and-play claim.

- **Supplementary zero-shot transfer experiments (R1, Q3):** Zero-shot transfer across agent counts is **not feasible in IMP**. To strengthen our validation instead, we add zero-shot resize experiments on AdversarialPursuit (Figs. 4(h–k)), where **SOPS maintains strong performance under population changes**.

- **Backbone vs learned selection ablation (R1, Q2):** To isolate the effect of learned selection from the exponential backbone, we add an ablation where the scorer is removed and edges are chosen by Euclidean distance with varying budget K (Fig. 5(b)). With the same backbone, **SOPS converges faster and more stably**, showing that adaptive neighbor selection is crucial for the gains.

- **Statistical validation (R2, W2; R3, Q1):** We compute AUC metrics for MAgent curves (Tabs. 4–5) and perform two-sample t-tests on AUC across three seeds for large-scale scenarios. **All p-values vs EC-O and EC-S are less than 0.05, confirming that large-scale gains are statistically significant.**

- **System-level profiling (R1, Q5):** Appendix C.3.5 reports wall-clock steps/s, peak GPU memory, and activation size versus agent count and realized out-degree on Battle.

- **Visualizations (R2, Q4):** Following the reviewers’ suggestions, we add visualizations of SOPS-induced dynamic subgraphs and spatial behaviors under zero-shot transfer (Figs. 9–10).

We believe these revisions address all main concerns raised during review and improve the clarity and completeness of the paper. We are grateful for the reviewers’ and AC’s guidance, which materially enhanced the manuscript.

---

### Meta-Review · Area_Chair_NNjo · 2026-01-06

**Summary:**

This paper proposes Sparse Topology Pairwise Scoring (SOPS), a communication framework for large-scale MARL. It devises a sparse communication topolocy called the exponential graph as a scalable backbone, which is parameterized by a pairwise scoring network to sample probabilistic subgraphs. The authors evaluate the method on MAgent and IMP benchmarks, claiming improved scalability, performance, and zero-shot transfer capabilities compared to baselines.

While the reviewers acknowledged the importance of the problem and the soundness of the scalability claims, the consensus leans towards rejection primarily due to concerns regarding the incremental nature of the contribution and the statistical significance of the performance gains.

**Reviewer Concerns:**

The rebuttal successfully resolved several technical concerns regarding system efficiency and mechanism analysis. The authors provided the requested system-level profiling data (memory, wall-clock time), clarifying the computational cost relative to baselines. They also added learned-graph diagnostics to explain the method's behavior during transfer and demonstrated the plug-and-play capability by extending experiments to QPLEX and SHAQ.

However, the concerns regarding novelty and statistical significance remain outstanding. Reviewer eN33 characterized the method as a refinement of the existing exponential topology paradigm rather than a fundamentally new approach. Furthermore, despite the experimental results consistently outperform other models, reviewers also noted that the statistical significance of these results appears limited. Finally, the absence of stronger, more current attention-based baselines also limits the assessment of the method's contribution relative to the SOTA.

**Reviewer Scores:**

Reviewer bpJA (4 -> 4): The reviewer requested system-level profiling and learned-graph diagnostics. While the authors provided this data, the fundamental concern regarding the marginal contribution would likely to remain.

Reviewer eN33 (4 -> 4): This reviewer questioned implementation details and the novelty of the approach. The rebuttal clarified the implementation, but the objection regarding limited novelty and marginal improvement remains unresolved.

Reviewer tRJ9 (6 -> 6): The major concern raised by the reviewer is about the statistical validation. To solve this, the authors haved provided detailed explanation and additional AUC metrics.

Reviewer tKLs (6 -> 6): The reviewer was positive but requested additional evidence on robustness and heterogeneity. The reviewer found the reponse satisfying and stated that he would maintain the initial position assessment.

---

### Decision · Program_Chairs · 2026-01-26

Reject